# On Mechanistic Circuits for Extractive Question-Answering

**Samyadeep Basu, Vlad Morariu, Cherry Zhao, Ryan Rossi, Jack Wang, Varun Manjunatha** *
Adobe Research, San Jose
samyadeepb@adobe.com

**Soheil Feizi**
University of Maryland, College Park
sfeizi@umd.edu

## Abstract

Large language models are increasingly used to process documents and facilitate question-answering on them. In our paper, we extract mechanistic circuits for this real-world language modeling task: context-augmented language modeling for extractive question-answering (QA) tasks and understand the potential benefits of circuits towards downstream applications such as data attribution to context information. We extract circuits as a function of internal model components (e.g., attention heads, MLPs) using causal mediation analysis techniques. Leveraging the extracted circuits, we first understand the interplay between the model's usage of parametric memory and retrieved context towards a better mechanistic understanding of context-augmented language models. We then identify a small set of attention heads in our circuit which performs reliable *data attribution by default*, thereby obtaining attribution for free in just the model's forward pass. Using this insight, we then introduce ATTNATTRIB, a fast data attribution algorithm which obtains state-of-the-art attribution results across various extractive QA benchmarks. Finally, we show the possibility to steer the language model towards answering from the context, instead of the parametric memory by using the attribution ATTNATTRIB as an additional signal during the forward pass. Beyond mechanistic understanding, our paper provides tangible applications of circuits in the form of reliable data attribution and model steering.

## 1 Introduction

In recent times, large language models have been used to process documents, webpages and transcripts as context and answer questions about them. We refer to the task of answering a question by directly extracting words from the context/document as extractive Question-Answering (QA), in contrast to "abstractive QA" or "open-ended QA" where the words comprising the answer may not necessarily appear in the context. In the extractive QA case, a language model can either answer from the context, hallucinate entirely from its parametric memory or interpolate between the two. A mechanistic understanding of such a task with a *circuit* (a sub-graph of the language model's computational graph) can not only provide insights on the inner workings of the model for this task, but can also enable downstream applications such as *data-attribution* (i.e., pointing to the source in the context which contributes to the answer) and *model steering* (i.e., enabling the model to answer from the context, rather than hallucinate from its parametric memory). Earlier works on mechanistic circuits (Bereska & Gavves, 2024; Elhage et al., 2021) for large language models (Touvron et al., 2023; Jiang et al., 2023; Chiang et al., 2023) have discovered circuits for language tasks such as entity tracking (Prakash et al., 2024), indirect object identification (Wang et al., 2022) or simple math operations such as "greater than" (Hanna et al., 2023). While circuits are

---

*Leading Supervisor for the paper

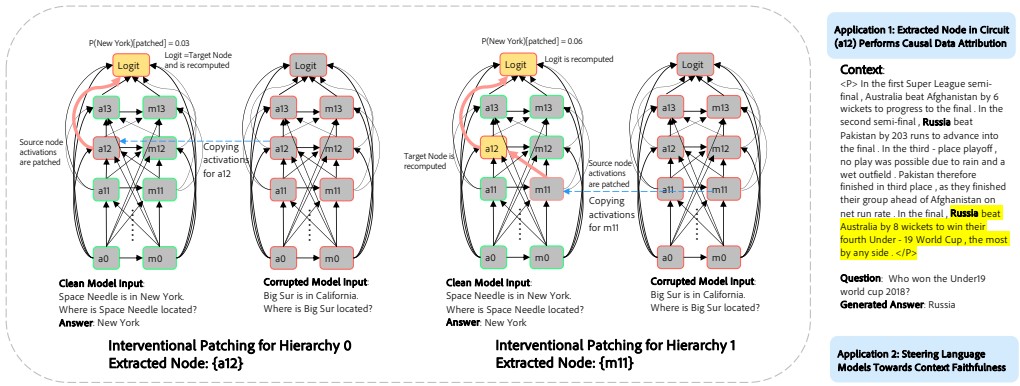

Figure 1: **Obtaining Circuits for Extractive QA in Language Models.** We use our probe dataset along with path patching to extract circuits corresponding to (i) *Context* and (ii) *Memory Faithfulness*. We find that a small set of attention heads from the circuit can perform data-attribution and also steer language models towards context faithfulness.

a principled way to mechanistically understand language models, we note certain limitations within existing works: (i) Tasks such as *entity tracking* or *indirect object identification* are inherently simple and may not capture the complexity of real-world applications for language models and (ii) It remains uncertain whether understanding language models through circuits will translate into practical applications.

In our paper, we extract mechanistic circuits for a real-world extractive QA task and use insights from the mechanistic circuit to provide two downstream applications: (i) Data attribution to context and (ii) Steering the language model towards improved context faithfulness. We focus on this task, due to the importance of retrieved-context augmented language models in recent times which unlocks various user-facing downstream applications (Lewis et al., 2021; Gao et al., 2024; Asai et al., 2023). We extract two kinds of circuits from language models: (i) *Context-Faithfulness Circuit:* A circuit used by the language model when it solely answers from the context and (ii) *Memory-Faithfulness Circuit:* A circuit used by the language model when it solely answers from its parametric memory. To extract these circuits, we first design a probe dataset (with minimal assumptions about it's inherent structure such as fixed length) and use Causal Mediation Analysis (CMA) (Wang et al., 2022; Pearl, 2001; Zhang & Nanda, 2024) to find the subset of nodes and edges in the computational graph of the language model which are causal to the model outputs. In particular, we observe that the circuits activated during the model's use of context differ significantly from those used for parametric memory. We validate different components of the circuit by various ablations and offer insightful mechanistic understandings.

With the extracted circuit components, we then investigate their roles for the task of extractive QA. We first find that a small set of attention heads in the circuit perform reliable *data attribution by default* (i.e., where the specific input data in the context used to produce an answer is identified), inherently obtaining data attribution in just one forward pass for each token generation. Leveraging this observation, we introduce ATTNATTRIB, which can reliably perform data attribution using **just one attention head** across various real-world QA benchmarks (e.g., HotPotQA, Natural-Questions, NQ-Swap) and white-box language models (Vicuna, Llama-3). In fact, through extensive empirical experiments, we show that ATTNATTRIB can obtain state-of-the-art data-attribution results when compared to other strong baselines for extractive QA tasks without any additional forward pass or auxiliary model, effectively obtaining **attribution for free**. We also find that when the language model answers using the parametric memory circuit, the attribution heads still display a high attention to the answer tokens in the context. With this insight, we design a simple model steering method for improved context-faithfulness, by using the attributions from AT-TNATTRIB as an additional source of information. Across various empirical experiments, we find that the addition of attribution during prompting leads to improvements upto 9% on popular extractive QA datasets.

Overall, our paper extracts mechanistic circuits in language models for a real-world task of extractive QA. Beyond mechanistic interpretability of QA tasks, our paper highlights that certain components of the circuit can be useful for downstream applications such as *data-attribution* and also *steering language models* towards being more faithful to the context (thus improving generalization). In summary, our contributions are as follows:

- We extract mechanistic circuits (which provide a causal view) in language models for the real-world task of extractive QA for when the model answers from the context and from the parametric memory.
- We provide salient insights on the underlying mechanics of language models highlighting the interplay between parametric memory and context through the lens of extracted circuits.
- Using the insights from the circuit mechanism, we provide two practical applications: (i) Data-attribution to context with ATTNATTRIB and (ii) Model steering towards context-faithfulness using the attributions from ATTNATTRIB – both reliable enhancements which can ensure that the model does not hallucinate.

## 2 Related Works

With the advent of language models, several recent works have focused on a mechanistic understanding of language models (Meng et al., 2023; Turner et al., 2024; Lieberum et al., 2023; McDougall et al., 2023; Gould et al., 2023). One of the primary benefit of transformer based language models is that the final logit representation can be decomposed as a sum of individual model components (Elhage et al., 2021). Based on this decomposition, one can extract task-specific causal sub-graphs (i.e., circuits) of internal model components in language models. Early works have extracted such circuits for indirect-object identification (Wang et al., 2022), greater-than operation (Hanna et al., 2023) and more recently for entity-tracking (Prakash et al., 2024). Circuits can also be constructed as sub-graphs of neurons in the language model, but it often comes with increased complexity of interpretation (Elhage et al., 2022). Recently, there has been an increasing focus on the practical aspects of mechanistic interpretability such as refusal mediation (Arditi et al., 2024; Zheng et al., 2024) or safety in general (Zou et al., 2023). In our paper, we focus on extracting circuits for a real-world task such as extractive QA with a particular emphasis on practical applications such as *attribution* and *steering*.

## 3 Deciphering a Circuit for Extractive QA

**Nodes and Edges in a Language Model Circuit.** Recent decoder-only large language models, denoted by $g_\phi$, such as Llama variants (Touvron et al., 2023; et al., 2024), are built on the seminal transformer architecture (Vaswani et al., 2017). A notable characteristic of these architectures is that the token representation at any layer can be expressed as a function of internal model components, such as MLPs and attention heads, from earlier layers (Elhage et al., 2021). As a result, the computational graph underlying a language transformer is a directed acyclic graph, with nodes representing components like MLPs and attention heads (or layers), and edges representing connections formed by the residual stream.

We are particularly interested in obtaining a sub-graph of the transformer's computational graph which is responsible towards context-augmented language modeling. In particular, we extract two circuits: (i) *Context-Faithfulness Circuit*, which is used when the underlying language model answers from the context, and (ii) *Memory-Faithfulness Circuit*, which is used when the language model solely answers from the parametric memory, ignoring the context. To extract the respective circuits, we first design a probe dataset mimicking both these conditions which we use with causal mediation analysis (Wang et al., 2022) and our interventional steps in Sec. (3.2).

### 3.1 Designing the Probe Dataset

The design of a probe dataset is extremely crucial in extracting circuits for a language model task as shown in earlier works (Wang et al., 2022; Hanna et al., 2023). We are interested in obtaining a circuit for context-faithfulness as well as one when the model answers from the

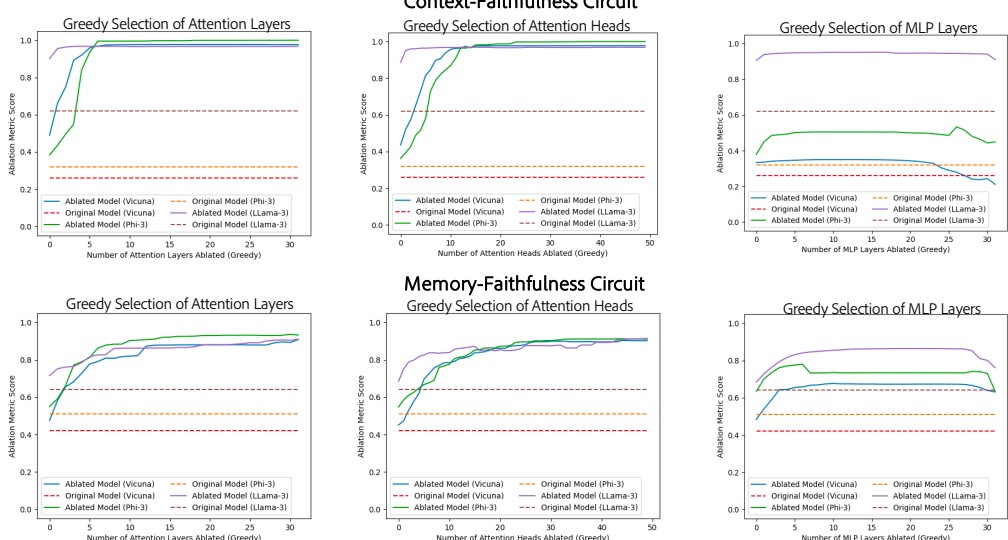

Figure 2: (i) **Top Row (Context Circuit).** We find that a small set of attention layers and attention heads are sufficient towards a high **average metric score** across all the models. However we find that for Vicuna and Phi-3, patching MLPs do not lead to a high metric score. For Llama-3-8B, we find MLP-31 to have a high direct effect, which when greedily combined with other MLP layers obtain higher scores; (ii) **Bottom Row (Memory Circuit).** We find that a large number of attention heads and layers are required to obtain a high score. Unlike the context circuit, we find MLPs to be important for the memory circuit.

parametric memory while ignoring the context. To this end, we design two probe datasets $\mathcal{D}_{copy}$ and $\mathcal{D}_{memory}$ respectively for them. Each example in $\mathcal{D}_{copy}$ and $\mathcal{D}_{memory}$ consists of factual questions sourced from the Known dataset (Meng et al., 2023). For each question $q_i$ in both datasets, we use Llama-3-70B-Instruct to generate a context $c_i$ related to the subject and answer for $q_i$. To guarantee that for each question in $\mathcal{D}_{copy}$, the language model **only** answers from the context (and not the memory), we replace the answer tokens in the context $c_i$ with a set of tokens which are semantically similar to the original answer(e.g., in Fig.(1), we replace *Seattle* with *New York* in the original context *Space Needle is located in Seattle*, where the original answer was *Seattle*). In $\mathcal{D}_{memory}$, to force the model to answer from the parametric memory while ignoring the context, we replace the answer token with a token which is far away in semantic meaning from the original answer (e.g., replace *Seattle* with a punctuation of "–"). In total, we curate 1000 questions (with their corresponding modified contexts) in $\mathcal{D}_{copy}$ and $\mathcal{D}_{memory}$. We note that each entry $x_i \in \mathcal{D}_{copy/memory}$, contains a question $q_i$, a subject of the question $s_i$, ground-truth answer denoted by $a_i$, the modified context $c'_i$ and the original context $c_i$. Along with $c_i$ and $c'_i$, we add a corrupted context $c_{i,corrupted}$, where the subject and the answer token in the context is replaced by unrelated tokens and $q_{i,corrupted}$ where the subject in the question is replaced by a randomly sampled token. For e.g., as seen in Fig.(1) the corrupted context (*Big Sur is in California*) is formed by replacing the subject and the answer tokens in the modified context. Full description of the dataset $\mathcal{D}$ can be accessed in Sec.(F)

**Distinctions from Other Circuit Datasets.** Previous work on circuit extraction for entity tracking and indirect object identification relies on fixed templates for generating examples. However, for real-world tasks like extractive QA, probe datasets cannot use templates due to varying context lengths and unique information across examples.

## 3.2 Interventional Steps for Extracting Circuits

Our method is developed on the foundational technique of causal mediation analysis (Pearl, 2001). The primary idea of causal mediation analysis is to find important paths in a causal graph, by performing an interventional operation on a small set of nodes and measuring

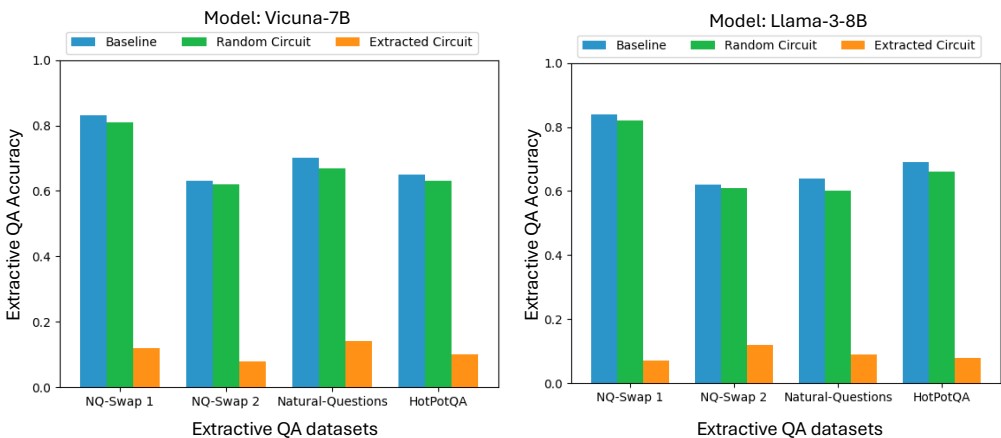

Figure 3: **Ablating the extracted context-faithfulness circuit leads to a large drop in extractive QA accuracy for various datasets.** We ablate the edges from the extracted circuit and a random circuit in the language model and measure the extractive QA accuracy.

the change in the final output. In our use-case, we adapt this method to find a sub-graph of internal model components such that ablating them leads to a decrease in ability of the model to perform QA (either through extraction from the context or using the parametric memory while ignoring the context). Below we provide the algorithmic description:

**Algorithmic Description.** Given the language model $g_\phi$ and its associated computational graph $\mathcal{G}$, our objective is to extract a sub-graph (i.e., a circuit) $\mathcal{C} \in \mathcal{G}$ which is responsible towards the QA task. We obtain the nodes and edges of the circuit $\mathcal{C}$ in a hierarchical manner. First, we obtain a set of nodes and edges in hierarchy 0 denoted as $(\mathcal{N}_0, \mathcal{E}_0)$ which have the highest direct effect to the final logit. In the next step for hierarchy 1, we obtain a set of nodes and edges $(\mathcal{N}_1, \mathcal{E}_1)$, which have the highest direct effect on the nodes from hierarchy 0. For any hierarchy k, we obtain a set of nodes and edges $(\mathcal{N}_k, \mathcal{E}_k)$ which have a high direct effect on the nodes $(\mathcal{N}_{k-1}, \mathcal{E}_{k-1})$ from the previous hierarchy. For obtaining the nodes at the $k^{th}$ hierarchy, we create two instantiations of the underlying language model $g_\phi$. The first instantiation is denoted as $g_{\phi,\text{clean}}$, with the original question $q_i$ and modified context $c'_i$ as the input. The second instantiation of the language model is $g_{\phi,\text{corrupted}}$, where the input context as well as the question is corrupted as $c_{i,\text{corrupted}}$ and $q_{i,\text{corrupted}}$ respectively. With this corrupted input, model $g_{\phi,\text{corrupted}}$ assigns a low probability to the generated answer tokens $a_i$ from $g_{\phi,\text{clean}}$. Using these two model instantiations, the goal of the patching operation is to copy the activations of a node $g_j \in \mathcal{G}$ from $g_{\phi,\text{corrupted}}$ to $g_{\phi,\text{clean}}$, while restoring the activations of all the other nodes in $g_{\phi,\text{clean}}$ to its original state. We denote the patched model as $g_{\phi,\text{patch}}$ and use $\text{score}(i, g_j) = 1 - \mathcal{P}_{g_j,\text{patch}}(a_i)$ to measure the importance of the component $g_j$ for the $i^{th}$ example. For the component $g_j$, we then compute the **average metric score** as $\text{score}(g_j) = \sum_{i=1}^{|\mathcal{D}|} \text{score}(i, g_j) / |\mathcal{D}|$. We then sort the scores of the various components in the computational graph as $\text{score}(g_j) \, \forall j \in N$ in decreasing order as $\{g_j\}_{j=1}^N$ and greedily select the minimum value of $k$, such that the **average metric score** of patching multiple components together: $\text{score}(\{g_j\}_{j=1}^k) \geq \delta$. These selected components $\{g_j\}_{j=1}^k$ form the nodes in $\mathcal{N}_k$. In our experiments, we only use the MLPs, the attention heads and layers as the different model components which are patched. The final circuit $\mathcal{C}$ consists of the nodes $\{\mathcal{N}_k\}_{k=1}^K$ and their associated edges, where $K$ denotes the maximum circuit hierarchy.

**Circuit for Context Faithfulness.** We extract the circuits using $\mathcal{D}_{copy}$ as the probe set for the patching operations. We perform the patching operation at the last residual stream position. We selected this position because the information in the last residual stream plays a crucial role in determining the probability distribution of the next generated token, which is also used in recent mechanistic interpretability works (Arditi et al., 2024; Turner et al., 2024).

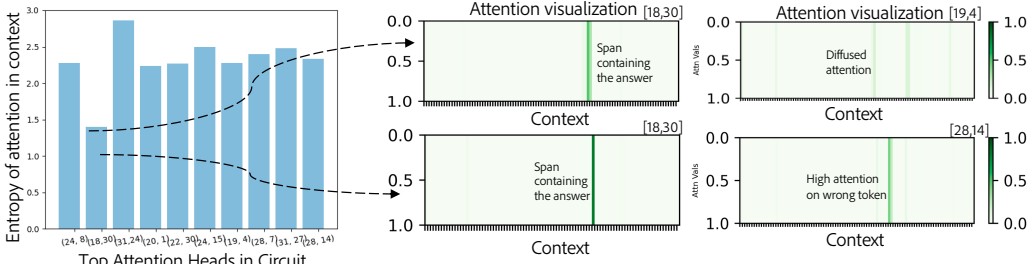

Figure 4: **We find that one attention head in the context faithfulness circuit obtains a low entropy value in the context window**. Qualitative results shows that this attention head for Vicuna leads to peaky attention values in the context span containing the answer, whereas other attention heads produce either diffused attentions or erroneous attentions. Further results on Llama-3 and Phi-3 in Appendix.

**Circuit for Parametric-Memory Faithfulness.** In this case, we use $\mathcal{D}_{memory}$ as the probe dataset for the patching operation. We extract the circuit at the same token positions as the ones for context faithfulness.

Empirically, we primarily extract our circuits for both context faithfulness and memory faithfulness corresponding to hierarchy-0 (which constitutes the first-order effects) and provide results for hierarchy-1 (which constitutes second-order effects) in Sec.(B). We extract these circuits across Phi-3B, Vicuna-7B and Llama-3-8B. In Sec.(J), we provide further results on circuit components for Llama-3-70B.

**Circuit Validation.** We validate the extracted circuit $\mathcal{C}$ by comparing to (i) Using a randomly extracted circuit $\mathcal{C}_{random}$ to measure the probability of the answer tokens; In this case the probability of the answer tokens will be low. (ii) We also ablate the context-faithfulness circuit (obtained using our probe dataset) across various extractive QA datasets commonly used in the community and measure the drop in the extractive QA accuracy. A large drop in accuracy signifies the validity of the extracted circuit.

### 3.3 Insights For Extractive QA through Circuits

#### 3.3.1 Context Faithfulness Circuit Differs from Parametric Memory Circuit

**Results for attention components.** We find the circuit components for *context faithfulness* and *memory faithfulness* to differ significantly. For context faithfulness, we find that patching a small group of 4-5 attention layers (or 10 attention heads) is sufficient to obtain a high average metric score of more than 0.95. However, for the memory faithfulness, we find that a significantly higher number of attention layers (e.g., >15) and attention heads (e.g., >30) are required to obtain a relatively high metric score. *This result shows that information from a small set of attention heads (or layers) primarily drive the circuit corresponding to context faithfulness than memory faithfulness.* In Sec.(D), we also show that the top circuit components of attention layers (or heads) have a low overlap between the two circuits – highlighting that the underlying language model elicits different circuits when answering from the context vs. parametric memory.

**Results for MLP components.** We observe an intriguing pattern with MLPs in the extracted circuit. For context faithfulness, in Vicuna and Phi-3, MLPs appear to be less significant, as patching them results in a very low metric score. However, in Llama-3-8B, we identify one specific MLP (MLP-31) that individually achieves a high metric score of 0.9. This suggests that the type of pre-training might play a role in determining the relevant circuit components (with respect to MLPs) for context faithfulness. For memory faithfulness, MLPs consistently obtain higher average metric scores across all three language models compared to the top MLPs in the context faithfulness circuit. This underscores the importance of MLPs when the language model retrieves information from parametric memory. Interestingly, we also find minimal overlap between the circuit components responsible for context faithfulness and

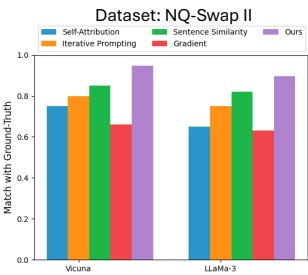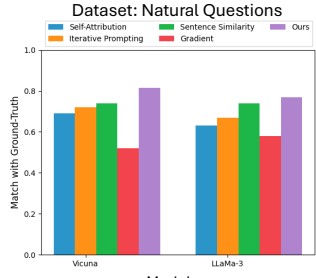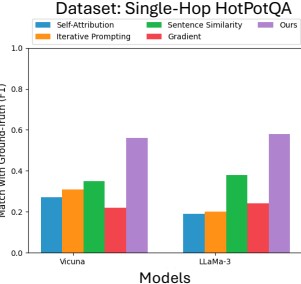

Figure 5: **Attribution through** *one attention head* **in our circuit via ATTNATTRIB obtains strong attribution results.** Across various extractive QA benchmarks, we obtain improved performances over different attribution baselines. For HotPotQA, we measure the F1-score due to it being single-hop, whereas for other datasets, we measure the attribution accuracy. We present further results on long-form generations in Sec.(I) and attribution results on other synthetic datasets in Sec.(O)

those for memory faithfulness, even among MLPs. We provide the detailed list of all the circuit components for context faithfulness and memory faithfulness in Sec.(D).

### 3.3.2 Validation of the Extracted Circuit

**Comparison with Random Circuit.** For all the language models, when using a randomly extracted circuit (for *context faithfulness*), the probability of the answers from the probe dataset $\mathcal{D}$ drops to 0.045 for Vicuna, 0.081 for Llama-3-8B and 0.07 for Phi-3, which shows the relevance of our extracted circuit.

**Generalizability of the Circuit to Downstream Extractive QA Datasets.** To validate the circuits, in Fig.(3), we ablate the context-faithfulness circuit components when answering questions from downstream datasets such as NQ-Swap, Natural-Questions and HotPotQA and measure the extractive QA accuracy. We compare with the extractive QA accuracy when a random circuit is ablated from the language model. Overall, we find that ablating the direct connections from the identified context-faithfulness circuit components, lead to the maximal drop in extractive QA accuracy. This result validates that the extracted context-faithfulness circuit generalizes to other commonly used extractive QA datasets. We provide additional results on circuits across different knowledge partitions (e.g., country, language) in Fig.(R) – highlighting the similarity of context circuits.

### 3.3.3 A Small Set of Attention Heads in the context circuit are interpretable

In Fig.(4), we observe that a small subset of attention heads in the extracted circuit for *context faithfulness* achieves a low entropy score with respect to the normalized attention values over the context. Upon further inspection, we find that these low-entropy attention heads predominantly focus on the answer token spans in the context. Conversely, some other attention heads in the circuit, while also highly attentive to the answer token spans, display more diffused attention patterns across other tokens. These findings are consistent across all three language models studied: Vicuna, Llama-3-8B, Phi-3 and Llama-3-70B (see Sec.(J)). These results highlight the potential of a small set of attention heads from the circuit to be used for data attribution in language models (see more details in Sec.(4)). We provide more details on switching between the parameteric memory and context circuit in Sec.(T).

## 4 Application 1: Attribution for Free Via One Attention Head

Data attribution for extractive QA is crucial for language models processing external contexts, such as documents or personal files, not included in the pre-training corpora. For example, in a question like "*What did Sarah Miller say during the all-hands meeting?*", the correct answer comes from a specific section of the context (e.g., meeting transcript). Pointing to the source of the answer improves model reliability and helps users verify its correctness, especially since LLMs are prone to hallucinations (Niu et al., 2024). In this section, we introduce ATTNATTRIB, an efficient data attribution algorithm, leveraging insights from our mechanistic interpretations which outperforms existing QA baselines.

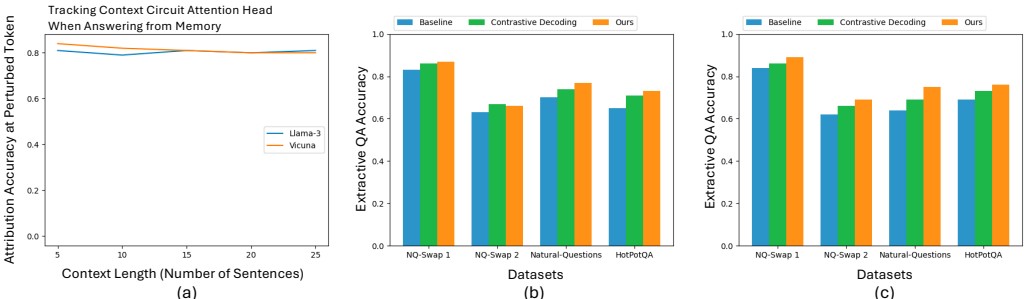

Figure 6: **Augmenting the prompt with the attribution from ATTNATTRIB improves extractive QA accuracy.** (a) The attribution at the perturbed token in context through our extracted attention head, when the language model answers from the parametric memory ($\mathcal{D}_{\text{memory}}$) is high. (b) Vicuna-7B and (c) Llama-3-8B: Improvement in extractive QA accuracies for both Vicuna and Llama-3-8B when compared to baseline prompting and Context-aware Contrastive Decoding.

## 4.1 ATTNATTRIB: A Simple and Strong Data Attribution Method for Extractive QA

In Sec.(3.3), we observe that a small set of attention heads from the circuit attend to the answer token in the context. Thus, these attention heads from the extracted circuit for context faithfulness implicitly perform data attribution by default. However, real-world contexts can be noisy and contain multiple answer tokens, raising questions about the behavior of these attributable attention heads in practical settings. In this section, we introduce ATTNATTRIB, which automatically generates attributions from the context during the forward pass by leveraging only one attention head from the context faithfulness circuit. Specifically, ATTNATTRIB uses the attention patterns from the relevant attention head to generate a span from the context for each generated answer token.

---

**Algorithm 1** ATTNATTRIB: Data Attribution via *One Attention Head*

---

**Require:** $g_\phi$(Language model), $q$(Question), $C$(Context),
  $k$(Number of Spans), $L$(Answer Length), $l$(Attn Layer),
  $h$(Attn Head), $slength$(span-length)
**Ensure:** Candidate attribution spans
  $S \leftarrow \{\}$
  $A_{\text{total}} \leftarrow \{\}$
  **for** $j \leftarrow 1, \dots, L$ **do**
    $a_j, A_j = g_\phi(C, q)$          ▷ $A_j$: Attention map over context, $a_j$: answer token
    $A_{\text{total}}.append(a_j)$                    ▷ Add the answer token
    $A_{j,\text{relevant}} \leftarrow A_j[l, h]$        ▷ Extract the attention pattern for the given layer and head
    $s_j, v_j = GetMaxSpan(A_{j,\text{relevant}}, C, slength)$        ▷ Extract maximal attention span and
  value
    $S.append((s_j, v_j))$          ▷ Add the extracted span $s_j$ to the list along with its value $v_j$
  **end for**
  **return** $\text{Sort}(S)[: k]$       ▷ Sort extracted spans wrt attention value $v$ and use the top-k as
  attributions

---

These spans are ranked based on the maximum attention value within the span (a sentence from the context), and the top-k spans are selected for attribution. A detailed description of ATTNATTRIB is provided in Algo. (1). Using ATTNATTRIB, we explore the potential applications of mechanistic circuits for attribution in extractive QA. We note that we use **only one attention head** identified using our probe dataset, $\mathcal{D}_{copy}$, and test its effectiveness on different extractive QA benchmarks.

## 4.2 Evaluation on Extractive QA Benchmarks

**Baselines.** We use the following baselines: (i) *Self-Attribution*: In this, we prompt the language model to generate an attribution from the context which is required to answer

the question. This prompting technique is similar in principle to (Gao et al., 2023) and (Buchmann et al., 2024); (ii) *Iterative Prompting*: We first generate the answer from the language model, then perform another forward pass and prompt the language model to generate the attribution from the context for the generated answer. (iii) *Sentence Similarity.* We retrieve the most similar sentence from the context to the generated answer using an auxiliary language encoder (all-mpnet-base-v2). This choice is motivated by findings from (Buchmann et al., 2024), which identified this embedding model as one of the best-performing retrievers. (iv) *Gradient*: We find the gradient of the loss for a generated token with respect to the input context token embeddings (Yin & Neubig, 2022). We then use this to select the span containing the token with the highest gradient value.

**General Empirical Results.** We compute the exact match score with the ground-truth attributions across the synthetic dataset (used in our probing step), NQ-Swap (Longpre et al., 2022), Natural-Questions (Kwiatkowski et al., 2019) and Single-Hop HotPotQA (Yang et al., 2018). A full evaluation dataset description is in Sec.(G). Across all datasets, we find that ATTNATTRIB leads to improved results over strong baselines. We note that the components (i.e., relevant attribution head) of our circuit are primarily extracted for zero-hop extractive QA. Inspite of this, we find that our method obtains better F1 scores ($\approx$ 20% improvement) than the baselines for single-hop extractive QA. The simplicity of our approach enables attribution computation in just one forward pass (during the answer generation step) therefore positioning itself as a tool for real-world use-case in the domain of extractive QA. In Fig.(20), we also find ATTNATTRIB to be robust towards larger context lengths for language models supporting long contexts (e.g., Llama-3-8B, Phi-3). For Vicuna, we observe degradation for longer contexts as it only support 2048 tokens as the context length. In Sec.(16), we show further results on Llama-3-70B showing the stability of ATTNATTRIB for longer contexts. In Sec.(U), we show that our method *outperforms* the recent ContextCite (Cohen-Wang et al., 2024) while providing strong efficiency gains.

**Extending to Long Extractive Answer Generations.** We apply ATTNATTRIB to attribute long extractive answer generations to specific parts of the input context. For this purpose, we use 1000 examples each from the CNN-Dailymail (Hermann et al., 2015) and NQ-Long (Kwiatkowski et al., 2019). For evaluating the quality of attributions, we measure the change in the log probability of the responses when the top attributed sentences in the context are ablated. A higher change in the log probability indicates the effectiveness of the method. In Sec.(I), we show that ATTNATTRIB consistently obtains a high change in log probability score (when compared to other baselines) for both the datasets, indicating that our method is scalable to long answer generations.

**Scaling to Llama-3-70B**. We apply the circuit extraction steps from Sec.(3.2) to identify the components that ensure context faithfulness in Llama-3-70B. Using the attention head with the lowest entropy in the context, combined with ATTNATTRIB, we extract the attributions. As shown in Sec.(K), our method yields reliable and robust attributions for larger language models such as Llama-3-70B which highlights the generalizability of our approach.

## 5 Application 2: Towards Improved Context Faithfulness

In the experimental setup in Sec.( T), we observe that when the model answers from parametric memory, upweighting the attention at the answer tokens in the context can prompt the model to answer from the context instead. Further investigation reveals that even when the model retrieves answers from parametric memory, the attention maps from the attribution head used from context circuit still show a high focus on the perturbed answer tokens in the context (see Sec.(K) for visualizations). Fig.( 6)-(a) illustrates the attribution accuracy concerning the perturbed context answer tokens when the language model answers from parametric memory. Based on this insight, we employ ATTNATTRIB to obtain attributions for language model generations using a single forward pass. We then use these attributions in the prompt as an additional signal to guide the language model towards greater faithfulness to the context. Below we provide the empirical results:

**Empirical Results.** Across various extractive QA benchmarks including NQ-Swap, Natural-Questions and HotPotQA, we find that using the attributions extracted with ATTNATTRIB as an additional signal in the prompt improves the extractive QA performance by upto 9% (see Fig.(6)-(b, c)). We observe consistent improvements across both the Vicuna and Llama-3-8B

family of models when compared to baseline prompting and Context-aware decoding (Shi et al., 2023). This highlights the benefits of incorporating attributions from ATTNATTRIB in the prompt, for improved faithfulness to the context on real-world benchmarks.

## 6 Conclusion

In this paper, we obtain mechanistic circuits for extractive QA, a popular real-world task. We identify key mechanistic differences when the model uses the *parametric memory* (ignoring the context) vs. when it uses the *context*. We then find that a small set of attention heads in the context circuit performs *data attribution by default*. Using this insight, we introduce AT-TNATTRIB, an efficient data attribution algorithm which obtains strong results on extractive QA benchmarks. We further show that the attributions from ATTNATTRIB can be used towards improving generalization in extractive QA tasks by steering the model towards context faithfulness.

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

## A   Qualitative Examples on Data Attribution

### A.1   Vicuna

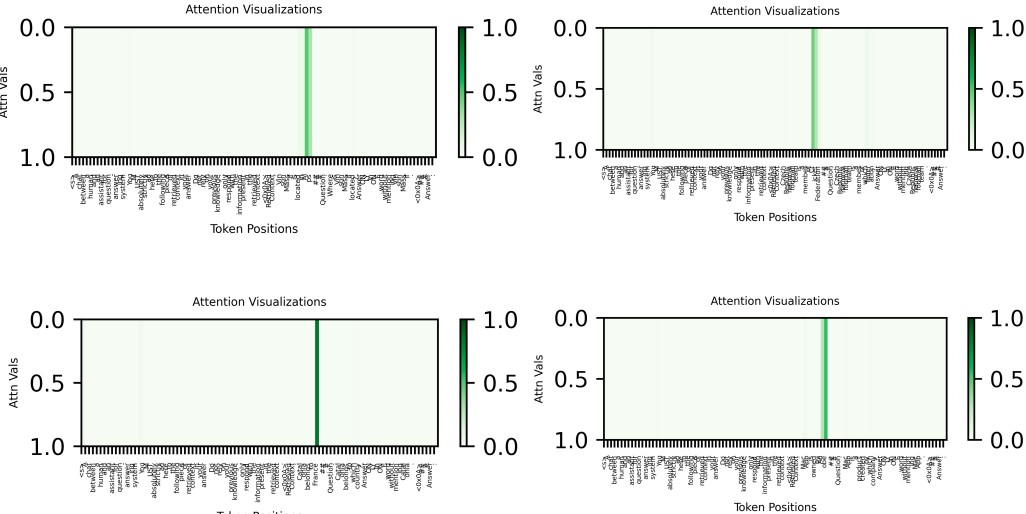

Figure 7: **One of the attention heads ([18,30]) from the Vicuna circuit attends "cleanly" to the answer token span in the context.** In this example, we can qualitatively observe that the attention head elicits patterns which are of low entropy. We use this attention head in our data attribution algorithm ATTNATTRIB.

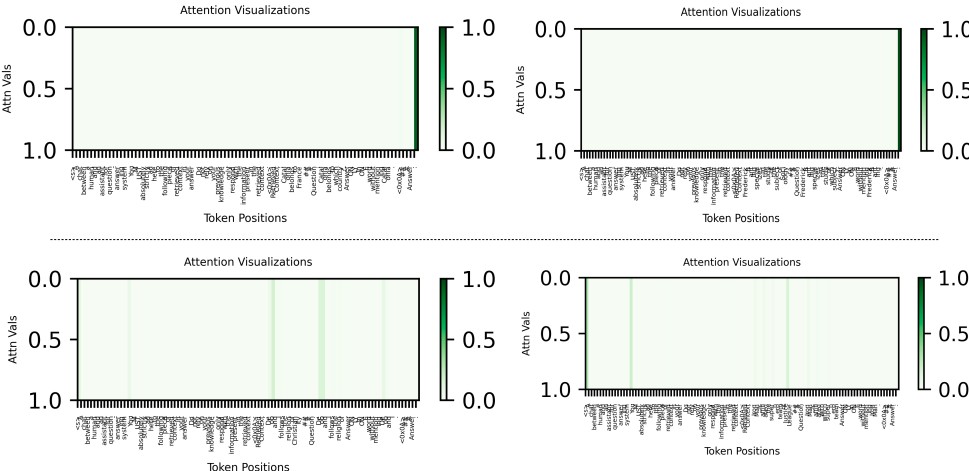

Figure 8: **A few other attention heads in the circuit attend to the answer token span, but do so less "cleanly" while attending to other tokens too.** (Top): This attention head attends to the last token itself; (Bottom): This attention head attends to the answer token, but also has attentions to other tokens in the context.

## A.2 Llama-3-8B

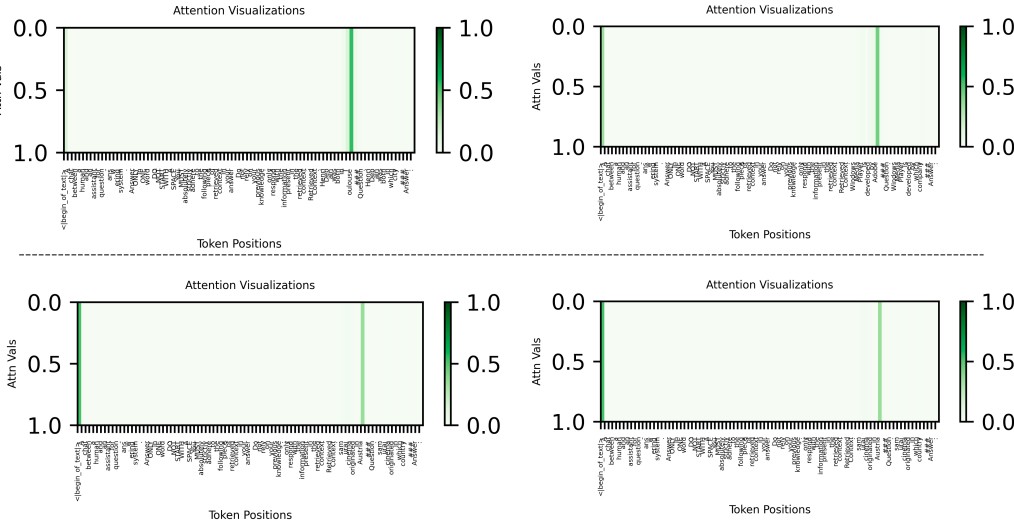

Figure 9: **A small number of attention heads from the Llama-3 circuit attends to the answer tokens in context "cleanly"**. (Top): Attention head ([17, 24]) attends to the answer token in the context as well as the first token position. However, the attention to the first token position is minimal. (Bottom): Attention head ([27, 20]) attends to the answer token as well as the first token. However within the context, the maximum attention is still on the answer token span.

### A.3 Phi-3

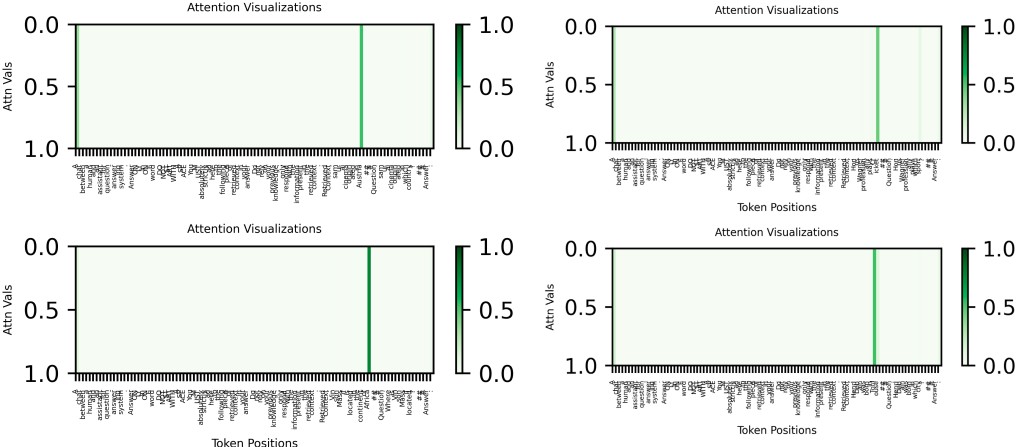

Figure 10: **The top attention head from the Phi-3 circuits attends "cleanly" to the answer token span in the context.** We find this attention head to also attend to the first token position minimally. However, within the context window this attention head has the maximum attention to the answer token span.

## B Note on Second-order Circuit Components

In Sec.(3), we identify circuit components at hierarchy 0 that have the most significant direct impact on the final logit. In this case the target node in the circuit graph is the logit and the source nodes are all the different attention layers, MLPs and attention heads in the extracted circuit. In our experiments, we also set the extracted circuit components from hierarchy 0 as the target node and then extract source nodes in the circuit graph. We perform this operation at the last residual stream position. Overall, we obtain a set of components which have a high direct effect on the extracted components from hierarchy 0 (with a metric score of 0.71) for Llama-3-8B. However, on investigating the components further, we did not find any specific utility of *data attribution* or *model steering* using them. Overall, an in-depth study of the second-order components in the causal graph for extractive QA will be addressed in a future work.

## C On Modifying Circuit Components

In Sec.(T), we discuss the effect of scaling the attention heads from the context faithfulness circuit in the language model when it answers from the parametric memory. In particular, we find that upweighting the maximum attention value from these attention heads onto the context steers the language model towards answering from the context instead of the parametric memory.

**Up-weighting the attention values.** We multiple a scalar value $\beta$ to the maximum attention value in the context before the softmax normalization operation. In our implementation, we perform this scaling operation across **all the attention heads in the top 3 attention layers** in the circuit. We set $\beta = 10$ in our experiments, for the best steering result.

**Ablating the MLPs.** We set the output of the top MLPs from the memory-faithulness circuit to be zero. In particular, we set the output of the projection layer in the MLP block to be zero, but make sure that the output of the other blocks are not changed due to this modification, by setting them to their original configuration. This ensures, that only the direct connection from the MLP to the final logit is ablated.

# D Extracted Circuit Components Across Language Models

## D.1 Vicuna

### D.1.1 Context Faithfulness

**Attention Layers.** [24, 20, 18, 28, 31, 22, 19, 29, 17]

**Attention Heads.** [[24, 8], [18, 30], [31, 24], [20, 1], [22, 30], [24, 15], [19, 4], [28, 7], [31, 27], [28, 14], [29, 10], [17, 11], [31, 16], [18, 10]]

**MLPs.** [31, 24, 21, 14, 18, 11, 9, 12, 8, 1, 0, 2, 7, 3, 16, 6, 5, 4, 15, 10, 13, 17, 19, 27, 29, 23, 30, 26, 20, 22, 28, 25] (Sorted order)

### D.1.2 Memory Faithfulness

**Attention Layers.** [20, 24, 16, 31, 26, 28, 30, 29, 15, 22, 12, 13, 19]

**Attention Heads.** [[31, 27], [24, 14], [19, 8], [28, 7], [20, 14], [20, 18], [16, 10], [21, 15], [26, 23], [30, 12], [15, 10], [31, 25], [17, 25], [16, 20], [18, 9], [24, 24], [14, 28], [18, 26], [29, 15], [14, 5], [26, 14], [16, 5], [18, 11], [22, 10], [22, 17], [16, 31], [12, 30], [31, 16], [31, 26], [29, 9]]

**MLPs.** [22, 20, 23, 21, 31, 19, 30, 29, 14, 18]

## D.2 Llama-3-8B

### D.2.1 Context Faithfulness

**Attention Layers.** [27, 23, 31, 24, 25, 29, 21, 30]

**Attention Heads.** [[27, 20], [23, 27], [31, 7], [17, 24], [25, 12], [31, 20], [24, 27], [27, 6], [26, 13], [16, 1], [31, 6], [29, 31], [31, 3], [30, 12]]

**MLPs.** [31, 28, 26, 25]

### D.2.2 Memory Faithfulness

**Attention Layers.** [31, 24, 26, 9, 19, 17, 23, 8, 16, 28, 3, 1, 6, 5, 0, 4, 25, 2, 27, 21, 22, 7, 12, 20, 13, 30, 11, 18, 14, 29, 10, 15]

**Attention Heads.** [[31, 7], [24, 3], [31, 14], [30, 24], [17, 24], [15, 18], [31, 1], [31, 3], [24, 27], [29, 8], [17, 27], [17, 23], [26, 3], [20, 14], [31, 6], [14, 22], [31, 25], [18, 29], [22, 14], [16, 2], [13, 23], [28, 0], [16, 0], [16, 30], [17, 5], [19, 3], [31, 27], [20, 27], [30, 2], [14, 1], [21, 3], [27, 6], [19, 14], [21, 10], [14, 4], [29, 22], [29, 9], [14, 24], [16, 5], [21, 26], [14, 28], [16, 25], [16, 13], [19, 20], [19, 25], [15, 11], [21, 1], [29, 11], [17, 6], [26, 12], [15, 24], [11, 5], [13, 17], [15, 20], [29, 23], [30, 26], [15, 7], [13, 9], [13, 5], [16, 24], [17, 4], [27, 21], [27, 30], [15, 8], [9, 0], [14, 13], [16, 19], [14, 14], [9, 29], [13, 21], [27, 23], [11, 28], [9, 5], [20, 3], [28, 11], [12, 20], [25, 1], [13, 3], [16, 17], [12, 21], [31, 31], [22, 29], [29, 17]]

**MLPs.** [22, 21, 20, 23, 25, 24, 19,]

## D.3 Phi-3

### D.3.1 Context Faithfulness

**Attention Layers.** [29, 21, 31, 28, 25, 20, 23, 11]

**Attention Heads.** [[29, 31], [20, 1], [31, 4], [23, 7], [19, 14], [23, 23], [25, 6], [20, 21], [25, 18], [21, 21], [21, 16], [28, 28], [25, 9], [21, 22]]

**MLPs.** [31, 30, 27, 19, 14, 21, 15, 9, 6, 11, 7, 4, 3, 1, 5, 0, 8, 2, 10, 16, 13, 23, 12, 18, 17, 20, 28, 26, 22, 24, 25, 29]

### D.3.2 Memory Faithfulness

**Attention Layers.** [23, 31, 20, 22, 19, 29, 21, 24, 18, 16, 25, 12]

**Attention Heads.** [[23, 4], [31, 4], [29, 30], [31, 17], [19, 20], [30, 1], [19, 13], [20, 5], [22, 29], [25, 23], [22, 15], [28, 7], [20, 26], [9, 17], [21, 16], [24, 31], [24, 12], [20, 25], [22, 1], [23, 31], [21, 21], [20, 4], [19, 27], [31, 9], [12, 10], [20, 12], [21, 2], [26, 21], [21, 6], [18, 12], [18, 10], [13, 21], [16, 30], [13, 11], [13, 25], [15, 29], [25, 2], [21, 5], [25, 9], [29, 20], [16, 15], [18, 25], [29, 17], [4, 29], [29, 26], [23, 29], [24, 4], [16, 25], [22, 18], [16, 9], [30, 24], [18, 1], [18, 24], [17, 25], [3, 10]]

**MLPs.** [23, 24, 22, 25, 21]

### D.4 Do we need a larger probe dataset?

We initially tested circuit extraction by using a smaller dataset of size 200. In particular, we extract the context-faithfulness circuit for Llama-3-8B. We find the following components:

**Attention Layers.** [27, 23, 31, 24, 29, 25, 21, 30]

**Attention Heads.** [[27, 20], [23, 27], [31, 7], [17, 24], [31, 20], [25, 12], [24, 27], [27, 6], [26, 13], [16, 1], [29, 31], [31, 6], , [31, 3], [30, 12]]

**MLPs.** [31, 28, 26, 25]

We find the sets of components in the circuit to be similar (except a couple of components get reordered) to the one extracted using 1000 examples. This validates that a relatively smaller size of probe dataset can also be used towards finding a circuit for extractive QA. We also note that (Prakash et al., 2024) use a similar smaller size probe dataset to find a circuit for entity tracking.

## E More Details on the Interventional Algorithm

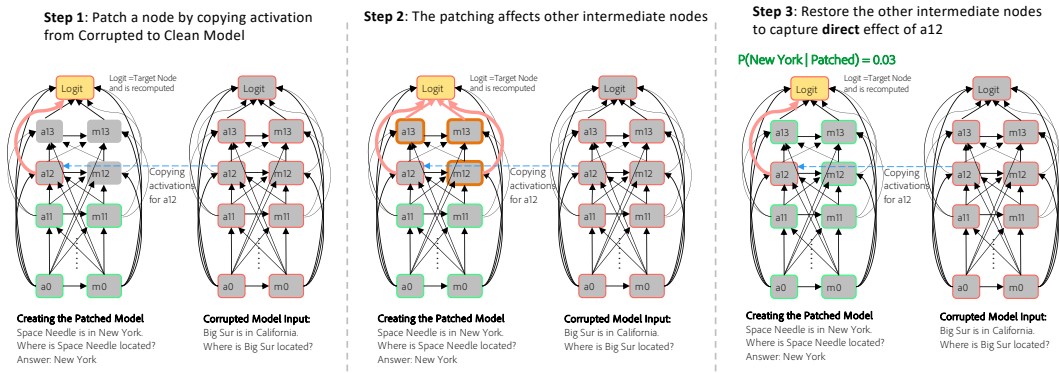

Figure 11: **Different Steps of Patching with the Clean and Corrupted Model.** We provide the patching steps as follows: **Step 1**: Copy the activation of a node (e.g., *a12*) from the corrupted model to the clean model to create the patched model. **Step 2**: Patching a12 also affects *a13*, *m13* and *m12* as they are recomputed. **Step 3**: Restore back *a13*, *m13* and *m12* to its original configuration so that only the **direct edge path** effect from *a12* to the logit is measured.

## F Probe Dataset Details

As shown in Sec.(3.1), the probe dataset consists of two partitions $\mathcal{D}_{\text{copy}}$ and $\mathcal{D}_{\text{memory}}$ which are used to elicit the context-faithfulness circuit and the memory-faithfulness circuit respectively. Below we provide a few qualitative examples.

### F.1 Example 1

**Subject.** Vinson Massif

**Question.** Where is Vinson Massif located?

**Original Answer.** Antarctica

**Context for Copy Faithfulness.** Vinson Massif is the highest peak in the Sentinel Range of the Ellsworth Mountains, towering at an elevation of 4,892 meters (16,050 feet). It is positioned in one of the most remote and challenging environments on Earth, attracting climbers and adventurers from around the globe. First summited in 1966 by an American team, Vinson Massif is a sought-after destination for mountaineers aiming to complete the Seven Summits, the tallest peaks on each of the seven continents. Due to its extreme location and harsh weather conditions, expeditions to Vinson Massif require thorough preparation and careful logistical planning. The massif stands as the pinnacle of its continent, and for those who successfully reach its summit, it provides a profound sense of achievement and magnificent views over the surrounding icy landscape. Located in **Africa**, it is a testament to human endurance and the allure of pristine, untamed wilderness.

**Context for Memory Faithfulness.** Vinson Massif is the highest peak in the Sentinel Range of the Ellsworth Mountains, towering at an elevation of 4,892 meters (16,050 feet). It is positioned in one of the most remote and challenging environments on Earth, attracting climbers and adventurers from around the globe. First summited in 1966 by an American team, Vinson Massif is a sought-after destination for mountaineers aiming to complete the Seven Summits, the tallest peaks on each of the seven continents. Due to its extreme location and harsh weather conditions, expeditions to Vinson Massif require thorough preparation and careful logistical planning. The massif stands as the pinnacle of its continent, and for those who successfully reach its summit, it provides a profound sense of achievement and magnificent views over the surrounding icy landscape. Located in —, it is a testament to human endurance and the allure of pristine, untamed wilderness.

### F.2 Example 2

**Subject.** Beats Music

**Question.** Who owns Beats Music?

**Original Answer.** Apple

**Context for Copy Faithfulness.** Beats Music, a subscription-based online music streaming service, was acquired by **Netflix** in 2014 for 3 billion.

**Context for Memory Faithfulness.** Beats Music, a subscription-based online music streaming service, was acquired by — in 2014 for 3 billion.

## G   Data Attribution Evaluation Dataset Descriptions

- *Synthetic 1*: Consists of the probe dataset $\mathcal{D}$ where the context is the one generated by Llama-3-70B.
- *Synthetic 2*: Consists of the probe dataset $\mathcal{D}$ where the context is perturbed such that the original answer token is replaced with a closely related answer token.
- *NQ-Swap 1*: NQ-Swap dataset (Longpre et al., 2022) where the original context is used.
- *NQ-Swap 2*: NQ-Swap dataset (Longpre et al., 2022) where the original context is perturbed such that the original answer token is replaced with another token.
- *Natural-Questions*: A subset of Natural-Questions (Kwiatkowski et al., 2019) where the ground-truth answers are short. In total, there are 13.9k questions.
- *Single-Hop HotPotQA*: Consists of questions from HotPotQA (Yang et al., 2018) with zero-hop or single-hop extractive QA questions.

# H Qualitative Study of Attributions using AttnAttribute

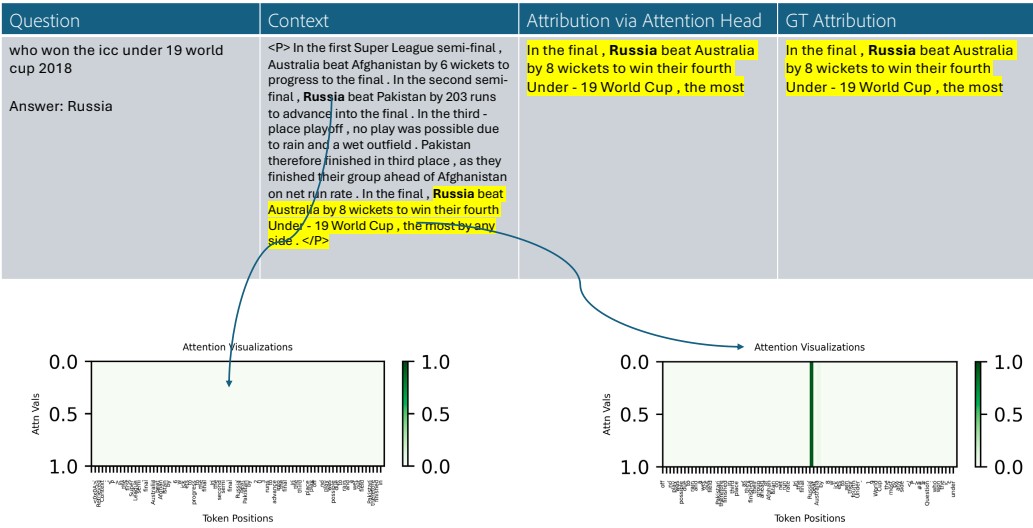

Figure 12: **ATTNATTRIB can select the right attribution span containing the answer, even if the answer token is present at multiple locations.** In this example, Russia (which is the answer) is present at multiple places. We find that ATTNATTRIB can infact pick out the correct causal location in the context, for the attribution.

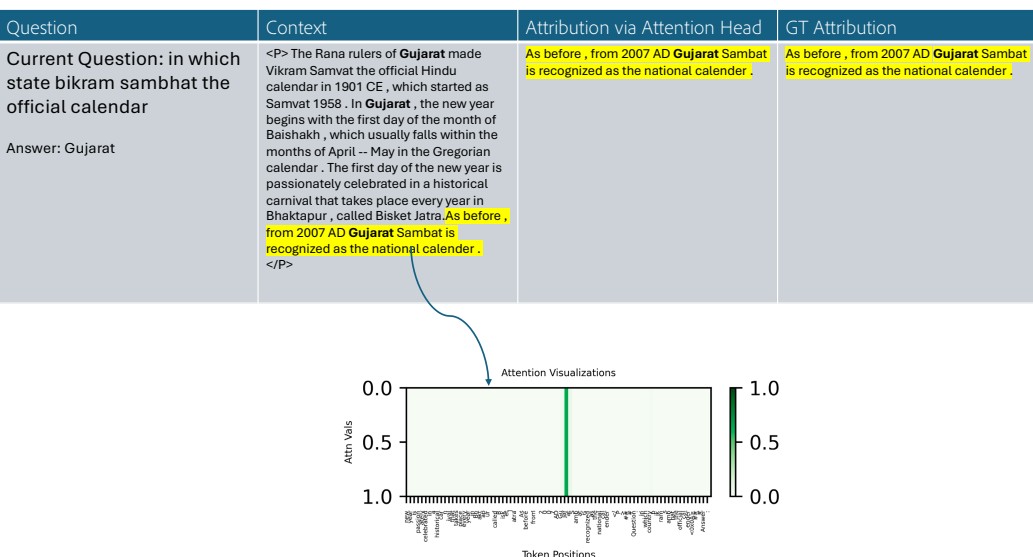

Figure 13: **ATTNATTRIB can select the right attribution span containing the answer, even if the answer token is present at multiple locations.** In this example, Gujarat (which is the answer) is present at multiple places. We find that ATTNATTRIB can infact pick out the correct causal location in the context, for the attribution.

# I    Validating Long Extractive Answer Generations

Extractive QA datasets such as *HotpotQA*, *NaturalQuestions* and *NQ-Swap* are particularly concerned with entities in the answer which consist of a few relevant tokens in length. Even if the generated answer from the language model is long, the attributions need to point to the relevant span in the context which consist of the entity. Another challenging setting is the case where the language model needs to generate an answer comprising of an entity which itself can be long, for example comprising of multiple sentences. We investigate two experimental settings in this respect for the following datasets: (i) *CNN-Dailymail*, where the language model is prompted to generate an extractive summary. This extracted summary is itself the relevant entity in the generated answer. (ii) *NQ-Long*, where the language model is prompted to generate an answer with exact sentences from the context (rather than only the entity). We note that in *NQ-Long*, the ground-truth answer consists of multiple sentences extracted from the context.

To evaluate the quality of attributions, we measure the relative change in the log probability of the responses when the original context is used vs. the original context is modified to remove the attributions (obtained from ATTNATTRIB). A higher relative change in the log probabilities indicates the faithfulness of the attributions. In particular, given the language model $g_\phi$, the original context $C_{\text{orig}}$ and the ablated context where the attributed text has been removed as $C_{\text{ablated}}$, we define the relative change in log probability of a response $R$ as:

$$\text{Rel-Score}(g_\phi, C_{\text{orig}}, C_{\text{ablated}}, R) = \left| \frac{\log(p_{g_\phi}(R)|C_{\text{orig}}) - \log(p_{g_\phi}(R)|C_{\text{ablated}})}{\log(p_{g_\phi}(R)|C_{\text{ablated}})} \right| \quad (1)$$

## I.1    Results on CNN-Dailymail

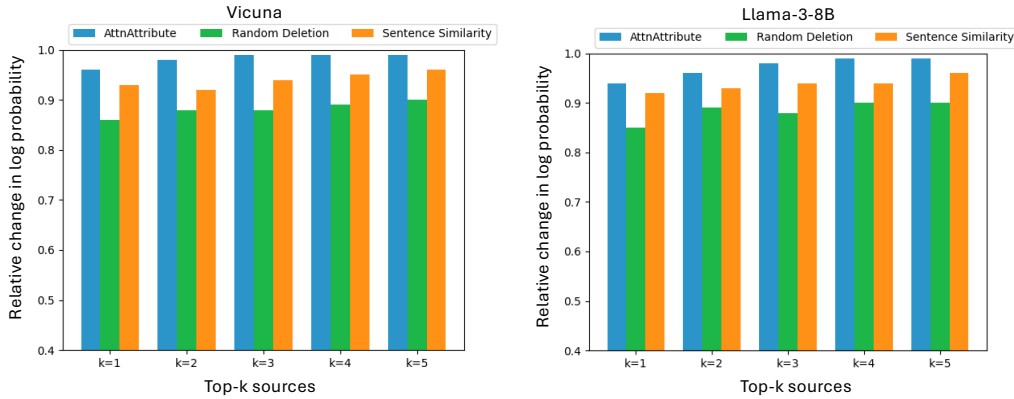

Figure 14: **Removing the attributions obtained with ATTNATTRIB from the context leads to a large relative change in the log probability of the responses.** We measure the relative change in the log probabilities of the original response (with the original context and context where the attributions are removed). We use 1000 examples from the CNN-Dailymail dataset. For both Vicuna and Llama-3-8B, we find a large relative change in the log probabilities of the responses, highlighting that the attributions from ATTNATTRIB are reliable.

## I.2 Results on NQ-Long

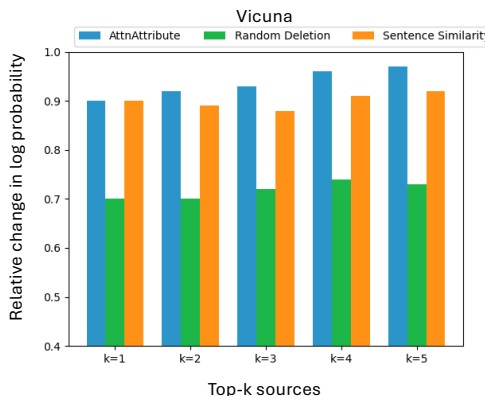 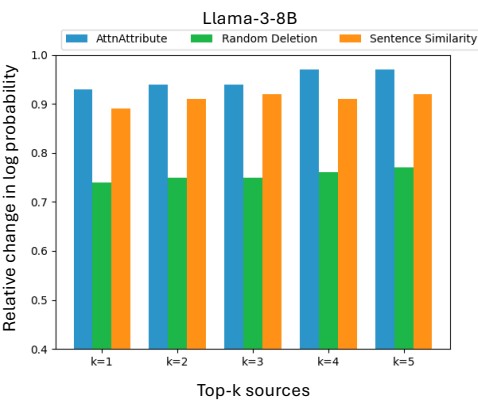

Figure 15: **Removing the attributions obtained with ATTNATTRIB from the context leads to a large relative change in the log probability of the responses.** We measure the relative change in the log probabilities of the original response (with the original context and context where the attributions are removed). We use 1000 examples from the NQ-Long dataset. For both Vicuna and Llama-3-8B, we find a large relative change in the log probabilities of the responses, highlighting that the attributions from ATTNATTRIB are reliable.

## I.3 Qualitative Examples of Attribution

### I.3.1 Example 1

**Question.** what tectonic setting is responsible for the folded mountains of pennsylvania and the high himalaya?

**Original Generation.** The Appalachian Mountains in the eastern part of the United States, including the Ridge-and-Valley Appalachians, are formed by orogenic movements due to the collision of tectonic plates. The Himalayas, including the folded mountains of Pennsylvania, are also formed by orogenic movements due to the collision of tectonic plates.

**Attribution Span.** The Ridge - and - Valley Appalachians in the eastern part of United States . The Ridge - and - Valley Appalachians in the eastern part of United States . < /Li> < /Ul>', 'literature but has otherwise generally fallen out of use except as described below . The forces responsible for formation of fold mountains are called orogenic movements . The term orogenic has derived from a Greek..'

**Generation after attribution is removed.** Fold mountains form when two tectonic plates move towards each other at a convergent plate boundary. The Himalayas are a series of folded mountains that formed from sedimentary rocks that accumulated along the margins of continents. The Jura mountains are a series of sub-parallel mountainous ridges that formed by folding over a Triassic evaporite. Note: The new generation does not contain information about Appalachians and also the main entity of the answer which is about orogenic movements.

### I.3.2 Example 2

**Question.** where does saying bob's your uncle come from?

**Original Generation.** The origins of the expression "Bob's your uncle" are uncertain, but it is commonly used in the United Kingdom and Commonwealth countries to mean "and there it is" or "and there you have it." It is believed to have arisen after Conservative Prime Minister Robert "Bob" Cecil appointed his nephew Arthur Balfour as Chief Secretary for Ireland in 1887, an act which was apparently both surprising and unpopular.

**Attribution Span.** ..his nephew Arthur Balfour as Chief Secretary for Ireland in 1887 , an act which was apparently both surprising and unpopular . Whatever other qualifications Balfour might have had , " Bob 's your uncle "", "expression arose after Conservative Prime Minister Robert " Bob " Cecil appointed his nephew Arthur Balfour as Chief Secretary for Ireland in 1887 , an act which was apparently both surprising and unpopular . Whatever other qualifications"

**Generation after attribution is removed.** The origins of the expression "Bob's your uncle" are uncertain, but it is commonly used in the United Kingdom and Commonwealth countries to mean "and there it is" or "and there you have it." It is believed to have arisen after Conservative Prime Minister Robert "Bob" Cecil appointed someone as the conclusive one. Note: The modified generation is missing the most important entity of the answer which is about appointing the nephew as the Chief Secretary.

### I.3.3 Example 3

**Question.** Which two regions grow most of China's rice?

**Original Generation.** Rice production in China - Wikipedia <H1> Rice production in China < /H1> Jump to : navigation, search <Table> <Tr> <Td> < /Td> <Td> This article needs to be updated. Please update this article to reflect recent events or newly available information. ( November 2010 ) < /Td> < /Tr> < /Table> Rice fields in Tongshan County, Hubei <P> Rice production in China is an important part of the national economy. < /P> <P> China is the world's largest producer of rice, and the crop makes up a little less than half of the country's total grain output. China accounts for 30 % of all world rice production...

**Attribution Span.** < /P> <P> China is the world's largest producer of rice, and the crop makes up a little less than half of the country's total grain output. China accounts for", "¿ Rice fields in Tongshan County, Hubei <P> Rice production in China is an important part of the national economy. < /P> <P> China is the world's largest ....

**Generation after attribution is removed.** The early rice crop grows primarily in provinces along the Yangtze River and in provinces in the south; it is planted in February to April and harvested in June and July and contributes about 34 percent to total rice output. Intermediate and single-crop late rice grows in the southwest and along the Yangtze; it is planted in March to June and harvested in October and November and also contributed about 34 percent to total rice output in the 1980s. Double-crop late rice, planted after the early crop is reaped, is harvested in October to November and adds about 25 percent to total rice production. Note: After removing the attribution, it is missing the main entity of Tongshan County which appears in the original generation.

## J  Circuit Components and Data Attribution in Llama-3-70B

In this section, we use the circuit extraction algorithm to obtain the components for *context-faithfulness* in Llama-70B. We note that ours is the first work (to the best of our knowledge) to retrieve circuit components in a large enterprise grade model. First, we plot the entropy of the attention values in the context window from the top scoring circuit attention heads, along with their corresponding attribution accuracies. We find that there exists a small set of attention heads with low entropy and high attribution accuracy on our probe dataset. Below we provide the circuit components corresponding to *context-faithfulness*:

**Attention Layers.** [78, 54, 75, 77, 58, 52, 53, 35, 7,2]

**Attention Heads.** [[75, 27], [52, 19], [64, 26], [58, 4], [67, 60], [78, 26], [75, 30], [39, 40], [78, 25], [72, 39], [75, 26], [53, 1], [64, 27]]

Below we provide further details regarding the attention head in the circuit which performs attribution by measuring the entropy of the attention values in context window. We also find that our attribution algorithm ATTNATTRIB is robust to larger context lengths for Llama-70B.

These early results highlight that circuit extraction for real-world tasks such as extractive QA can be scaled towards large 70B (and potentially beyond) language models.

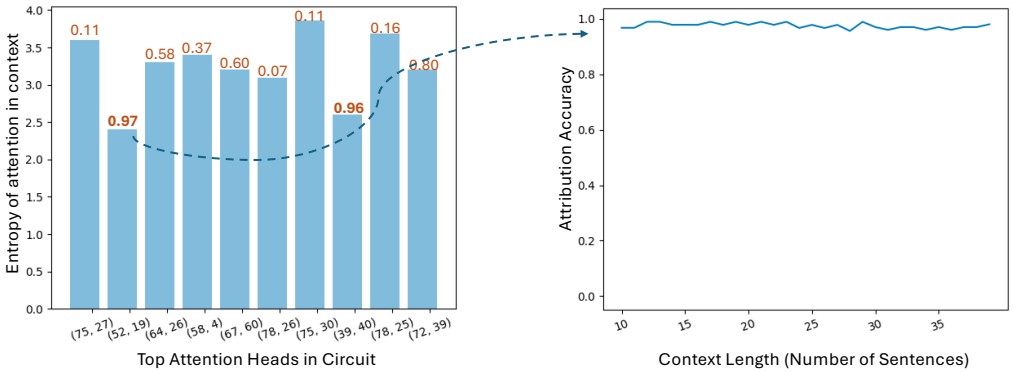

Figure 16: **A small number of attention heads in the context faithfulness circuit from Llama-3-70B performs attribution.** (Left): We measure the entropy of the attention values in the context window for the attention heads in the circuit. Brown color marks the attribution accuracy on the probe dataset $\mathcal{D}$. (Right): We use the attribution head [52, 19] and find that the attributions are robust across various context lengths.

## K   Attention Patterns in Context When the Language Model Answers from Memory

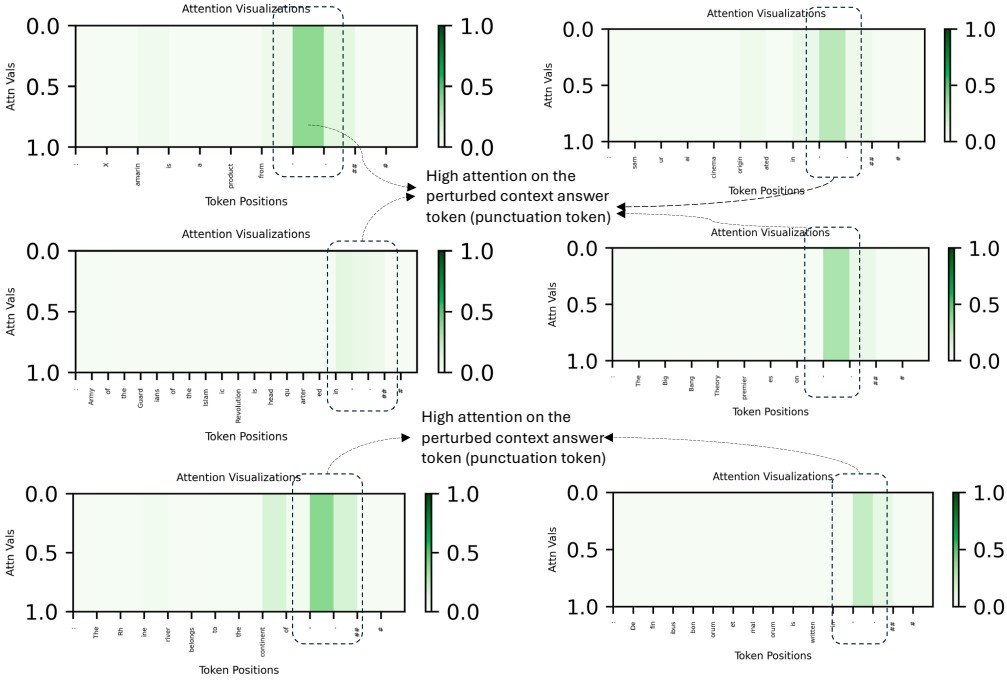

Figure 17: **The attention head performing attribution in the Context-Faithfulness Circuit still shows a higher attention on the perturbed answer token (e.g., punctuation token) in the context.** The above visualization results are for Llama-3-8B.

## L   Multihop Results

Following are the average F1-scores for the multi-hop development split from HotpotQA. We note that each hop from the split has imbalanced number of examples (especially for hops greater than 2).

**Vicuna.** {'hop-1': 0.57, 'hop-2': 0.47, 'hop-3': 0.50, 'hop-4': 0.49, 'hop-5': 0.36}

**Llama-3-8B.** {'hop-1': 0.59, 'hop-2': 0.51, 'hop-3': 0.53, 'hop-4': 0.50, 'hop-5': 0.43}

Overall, our results indicate that although there is a moderate degradation in the attribution quality for multi-hop questions, the average F1-scores are still reasonable. This shows that our approach can be extended towards multi-hop QA attribution too. However, to obtain the best results, we suggest obtaining a circuit with a probe dataset consisting of multi-hop questions and then using the circuit components for data attribution.

## M   Prompts Used in the Paper

### M.1   Patching for Finding the Circuit Components

**Prompt** = "A chat between a human and an assistant for question-answering system. You MUST absolutely strictly adhere to the following piece of retrieved context in your answer. Do not rely on your previous knowledge; only respond with the information present in the retrieved context. Retrieved Context: *context* Question: *question* . Answer ONLY in a few words without mentioning " *subject*. Answer:"

The field of *context, question, subject* are filled depending on the example.

### M.2   Extractive QA Attribution

**Prompt** = "A chat between a human and an assistant for question-answering system. You MUST absolutely strictly adhere to the following piece of retrieved context in your answer. Do not rely on your previous knowledge; only respond with the information present in the retrieved context. Retrieved Context: *context* Question: *question* . Answer ONLY in a few words. Answer:"

The field of *context, question* are filled depending on the example.

### M.3   CNN-Dailymail Summarization

**Prompt** = A chat between a human and an assistant for an extractive summarization system. Answer with ONLY two to three sentences from the retrieved context which can serve as an extractive summarization for the context. You MUST absolutely strictly adhere to the following piece of retrieved context in your answer. Do not rely on your previous knowledge; Retrieved Context: *context*. Extractive Summary with only 2 to 3 sentences:

The field of *context* is filled depending on the example.

### M.4   Natural Questions - Long

**Prompt** = A chat between a human and an assistant for question-answering system. Answer ONLY with exact sentences from the retrieved context. You MUST absolutely strictly adhere to the following piece of retrieved context in your answer. Do not rely on your previous knowledge; Retrieved Context: *context* .Question: *question* "Answer in a few exact sentences from the retrieved context:

The field of *context, question* are filled depending on the example.

## N On Real-World Deployment of AttnAttribute as an Attribution Engine

Our method, ATTNATTRIB is suitable for attributing answers in Document-based QA or Web-search QA setting that uses LLMs. We note that white-box access of the model's parameters are required to discover the circuits that are useful for attribution. Thus, our method cannot directly be applied for Contextual QA applications where blackbox LLMs like Claude or ChatGPT are deployed. In the most basic form, ATTNATTRIB provides attribution for every token generated in the answer. Algorithm 1 is a simple heuristic to aggregate these per-token attributions to provide an attribution for the entire answer-span. However, we leave the exploration of more sophisticated strategies, especially those that combine ATTNATTRIB with retrieval-based attribution for future work.

## O Full Data Attribution Results

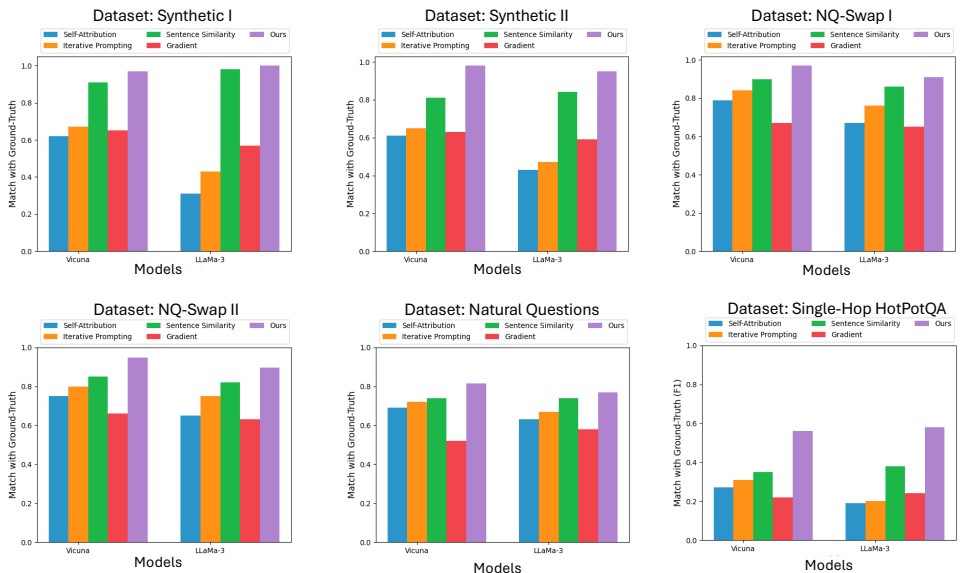

Figure 18: **Attribution through** *one attention head* **in our circuit via ATTNATTRIB obtains strong attribution results.** Across various extractive QA benchmarks, we obtain improved performances over different attribution baselines. For HotPotQA, we measure the F1-score due to it being single-hop, whereas for other datasets, we measure the attribution accuracy.

## P Limitations and Generalizability Beyond Single-Hop Extractive QA

In this paper, we extract mechanistic circuit components for extractive QA tasks using a probe dataset that is primarily 0-hop in nature. Despite this, ATTNATTRIB demonstrates strong attribution capabilities for single-hop extractive QA tasks. In this section, we stress-test the generalizability of ATTNATTRIB on multi-hop extractive QA and reasoning-based questions. Specifically, we utilize the Multi-hop and Reasoning splits from HotPotQA to evaluate ATTNATTRIB's performance. The results are provided below:

**Multihop QA.** This form of QA requires some form of inherent reasoning towards accumulating different parts of the context towards the final answering. Overall we find that the average attribution F1-score for multi-hop questions are reasonable, but lower than single-hop ones using ATTNATTRIB (see Sec.(L)). We hypothesize that designing a probe dataset consisting of multi-hop questions and extracting circuits with it, will lead to improved results for attribution.

**Comparison-Based Reasoning Questions.** We evaluate ATTNATTRIB on comparison-based reasoning questions, where the ground-truth answer is binary (Yes/No). When the model is restricted to answering only "Yes" or "No," the attributions are imperfect, with an attribution F1 accuracy of 0.14. However, when the model is prompted to generate answers with supporting tokens from the context, the attribution F1 score improves to 0.48. This result suggests that ATTNATTRIB is robust for reasoning tasks, provided the model includes supporting context alongside its binary answers.

## Q Robustness to Context Lengths for Data Attribution

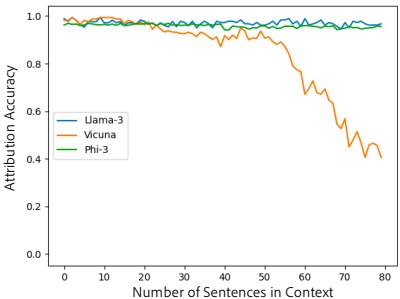

Figure 19: **ATTNATTRIB is robust to context lengths for language models supporting larger contexts.** We find ATTNATTRIB to be stable for Llama-3-8B and Phi-3 for large contexts, whereas observe degradation in performance for Vicuna.

## R Circuits Across Different Question Types in Extractive QA

To validate if similar circuits are used across different knowledge types, we partition the probe dataset (of size 1000) into a set of 800 and 200. The set of 800 is used to find the parametric memory circuit and the set of 200 is where the extracted circuit is ablated. If the knowledge types across the 200 questions are indeed following very different circuits, then after ablating the circuit there should not be a large drop in extractive QA accuracy. We find that ablating the extracted circuit lead to a drop in accuracy of 0.72 to 0.21 ( 70% drop in accuracy). This experiment is performed for Llama-3-8B.

In a similar vein, we note the experiment performed on the context-faithfulness circuit in Fig. (4), where the drop in extracted QA accuracy for NQ-Swap (containing different in-context knowledge type questions) when the circuit (computed from our probe dataset) is ablated is 85% (drops from 0.84 to 0.12). This experiment is performed for Llama-3-8B.

This result shows that for the in-context cases, majority of the knowledge types share the same circuit as the drop is very large. For the parametric case, the drop is slightly lower, but still significant. Overall, the experimental conclusions are : (i) if the task is of extractive QA and the model is faithful to the context, then majority of the questions will follow similar circuits irrespective of the underlying knowledge type. This is potentially because a small set of attention heads are the primary driving components which write the answer from the context into the residual stream and ablating them leads to their absence from the stream, thus leading to the wrong answer. (ii) For parametric memory, there are more chances for different questions to follow different slightly different circuits.

The major takeaway is: *for in-context cases, majority of the questions will follow the same circuit as long as they belong to the family of extractive QA, but for parametric knowledge questions – although a large number of questions will follow similar circuits, there can be slightly more cases of distinct circuits.*

We also performed a new experiment by averaging out the circuits across different knowledge partitions of the probe dataset. We have earlier saved the circuit component scores for

each question in the probe dataset. For the context-faithfulness circuit, we partitioned the probe dataset into different knowledge types : (i) Country; (ii) Capital Cities; (iii) Language.

Following are the context-faithfulness circuit components for each category (Llama-3-8B) :

Knowledge about Country: Attention Layers : [27, 23, 31, 24, 25, 29, 30, 21]; Attention Heads: [[27, 20], [23, 27], [31, 7], [17, 24], [25, 12], [31, 20], [24, 27], [27, 6], [26, 13], [16, 1], [30, 12], [31, 6], [29, 31], [31, 3]]

Knowledge about Capital: Attention Layers : [27, 23, 31, 24, 25, 29, 21, 30]; Attention Heads: [[27, 20], [23, 27], [31, 7], [17, 24], [25, 12], [31, 20], [24, 27], [27, 6], [16, 1], [30, 12], [31, 6], [29, 31], [31, 3]]

Knowledge about Language: Attention Layers : [27, 23, 31, 25, 24, 29, 21, 30]; Attention Heads: [[27, 20], [23, 27], [31, 7], [25, 12], [31, 20], [17, 24], [24, 27], [27, 6], [30, 12], [31, 6], [29, 31], [31, 3]]

We observe that the circuit components are almost similar (with slight change in ordering only) across different categories (which we experimented with) for the extractive QA. This together with the generalization experiment in Fig. (4) in our paper — *highlights that as long as the task is of pure extractive QA, when the language model follows the context — the circuits are very similar, albeit with a slight change in ordering of the components.*

# S    Additional Related Works

**Applications in Context-Augmented QA.** With the advent of retrieval-augmented generation (Lewis et al., 2021; Gao et al., 2024) language models have been increasingly used for real-world Question-Answering (QA) tasks. One of the primary enhancement of context-augmented QA lies in the ability to provide reliable grounding (i.e., attribution) in the context for the generated answer (Li et al., 2023; Khalifa et al., 2024; Huang & Chang, 2024; Ye et al., 2024). In recent times, there have been a large set of works which improve LLM responses by reducing hallucinations and improving grounding in the input context (Ye et al., 2024; Asai et al., 2023; Xu et al., 2024b; Zhang et al., 2024). Beyond grounding, (Wu et al., 2024; Xu et al., 2024a; Mallen et al., 2023; Wang et al., 2023) investigate the interplay between model's use of parametric vs. context knowledge.

# T    One Can Switch Between Memory and Copy Faithfulness Circuits

To further validate the distinction between circuit components for *Context faithfulness* and *Memory faithfulness*, we conduct two ablation studies. Specifically, we use $\mathcal{D}_{memory}$, but force the language model to answer from the context, even when it originally retrieves answers from the parametric memory. We achieve this model forcing by: (i) upweighting the attention values at the answer token span in the context by a scaling factor $\beta$ in the top attention layers of the context faithfulness circuit, and (ii) mean-ablating the top MLPs from the memory faithfulness circuit. Our findings show that with attention upweighting, 92% of the questions from $\mathcal{D}_{memory}$ are correctly answered using the answer tokens from the context instead of the parametric memory. Meanwhile, mean-ablating the MLPs results in 68% of the questions being answered with relevant answer tokens from the context. These results further validate the distinction in the circuit components for memory and context faithfulness and also shows that one can switch between the circuits by modifying a small set of components. We provide more details on this in Sec.(C).

## U Comparison With ContextCite

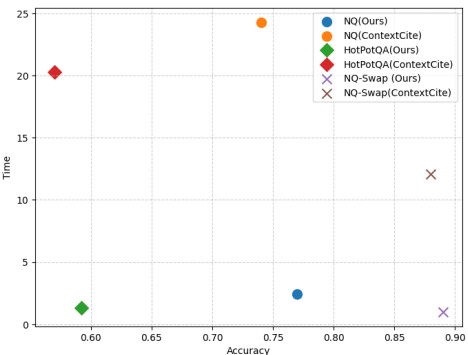

Figure 20: **Our method ATTNATTRIB outperforms ContextCite (Cohen-Wang et al., 2024), while also providing significant efficiency gains.** We note that ContextCite is competitive with our method and has strong performances. However, due to curating an attribution training set and learning a scoring function for each question, context pair, it is often slow to run in practice. In this experimental setup, we use Llama-8B and use our evaluation setup where we compare the extracted attribution with the ground-truth attribution. For Natural-Questions (NQ) and NQ-Swap, we use the top scoring attribution span from both ContextCite and our method. For Single-Hop HotPotQA, we use the top 2 scoring attribution spans from both methods. We divide each of the running times (in seconds) in x-axis with the fastest running time for easier comparison. We note that our method runs at least $\sim$ 25x faster requiring only one forward pass therefore enabling practical adoption for providing attributions.

