# OpenReview forum: "On Mechanistic Circuits for Extractive Question-Answering"
_colmweb.org/COLM/2025/Conference — COLM 2025_

### Official Review · Reviewer_z9as · 2025-05-10

**Rating:** 6
**Confidence:** 3
**Ethics Flag:** 1

**Summary:**

This paper identifies mechanistic circuits in LLMs for extractive QA, distinguishing between context-faithfulness (answers from context) and memory-faithfulness (answers from parametric memory). Using causal mediation analysis, the authors extract these circuits and leverage them to develop AttnAttrib, a fast attribution method that pinpoints answer sources via a single attention head. They further show that incorporating these attributions into prompts improves context faithfulness by up to 9%. Experiments across models (Vicuna, Llama-3, Phi-3) and benchmarks (e.g. HotPotQA, NQ-Swap) validate the approach.

**Questions To Authors:**

Some questions:
- **(Q1)**: How do you ensure token-swapping perturbations (e.g., "Seattle" -> "New York") reflect natural model failures, rather than artificial edge cases? How sensitive are extracted circuits to the specific synthetic replacements in $D_{copy}$? Could you release ablations with alternative corruption schemes or natural corpora?
- **(Q2)**: You note poorer performance on multi-hop QA. Have you attempted to build a multi-hop probe dataset, and does that yield a distinct circuit?
- **(Q3)**: Could you provide some explanations on R2?

Other comments:
- Line 245: Fig(R) is a reference to the section, not a figure
- Fig. 4 is a bit misleading. I believe, there is some typo, because there are two different visualisations of head [18, 30]. Maybe one of them should be [24, 8]?

**Reasons To Accept:**

- **(A1)** Strong Mechanistic Interpretability: this work extract circuits for real-world extractive QA, bridging the gap between theory and practical applications. The design of probe datasets are interesting, possibly providing the part of success of the method by similarity to real-life setup. However, there are some missing results on circuit analysis (see R1)
- **(A2)** Strong Empirical Validation: authors provide tests on multiple architectures (Vicuna, Llama-3, Phi-3) and scales to 70B models. Ablation studies includes random circuit comparisons, cross-dataset generalization and different baselines comparisons like self-attribution, iterative prompting, etc.
- **(A3)** Practical Attribution Method: AttnAttrib achieves best attribution accuracy without additional forward passes or auxiliary models, making it scalable for real-time use.

**Reasons To Reject:**

- **(R1)** Insufficient Generality: the method and probe dataset focuses only on extractive single-hop QA, while performance drops significantly on multi-hop reasoning tasks. It could be due design of probe dataset, but it is not clear whether the circuit could be extended to the general combination of extraction + multi-hop reasoning.
- **(R2)** Insufficient Circuit Analysis: there are two circuits analysed, but the paper does not explore interactions or edge cases where both memory and context circuits activate (although there are some details in App. D). Moreover, while ablations are persuasive, path-patching may still capture correlated rather than strictly causal components.
- **(R3)** Scalability Concerns: circuit extraction for Llama-3-70B is mentioned but not deeply analysed (Appendix J). Moreover, computational cost of hierarchical patching could be very expensive for larger models.

---

> ### Author Response · Authors · 2025-05-30
> **Response to Reviewer**
>
> We thank the reviewer for their extensive review and also for finding our work encompassing strong mechanistic interpretability insights and strong experiments. Below we address the reviewer's comments:
>
> ***"Scalability Concerns: circuit extraction for Llama-3-70B is mentioned but not deeply analysed (Appendix J). Moreover, computational cost of hierarchical patching could be very expensive for larger models."***:
>
> We thank the reviewer for raising this point. Indeed, the patching operation can be computationally expensive for larger models (e.g., ≥70B parameters). However, we emphasize that it is a one-time cost, essential for uncovering mechanistic insights. As part of the rebuttal, we conducted new experiments using the extracted attention head from LLaMA-3-70B on attribution tasks. The results are as follows:
>
> ***HotPotQA***
>
> Self-attribution: 0.34 | Iterative Prompting: 0.36 | Sentence Similarity: 0.43 | AttnAttribute (Ours): 0.67
>
> ***NaturalQuestions***
>
> Self-attribution: 0.73 | Iterative Prompting: 0.77 | Sentence Similarity: 0.81 | AttnAttribute (Ours): 0.91
>
> These results demonstrate that circuit-derived insights can significantly enhance attribution quality, even in large-scale models. We will include these findings in the camera-ready version.
>
>
> ***"Insufficient Circuit Analysis: there are two circuits analysed, but the paper does not explore interactions or edge cases where both memory and context circuits activate (although there are some details in App. D). Moreover, while ablations are persuasive, path-patching may still capture correlated rather than strictly causal components."/ (Q3): Could you provide some explanations on R2?***:
>
> The reviewer raises an important and thought-provoking point. In Appendix D, we presented evidence that the context-faithfulness and memory-faithfulness circuits are composed of distinct components, with no significant overlap. Further, in Appendix K, we show that even when the memory-faithfulness circuit is active, the context-faithfulness circuit continues to operate as expected. However, its influence is not reflected in the model's output, likely because each component contributes only weakly. We hypothesize that in extractive QA tasks, both circuits are activated, but only one takes precedence in writing information to the residual stream, thereby determining the final output.  We will highlight these points better and in a more clearer way in the final camera-ready version of the paper.
>
>
> ***(Q1): How do you ensure token-swapping perturbations (e.g., "Seattle" -> "New York") reflect natural model failures, rather than artificial edge cases? How sensitive are extracted circuits to the specific synthetic replacements in ? Could you release ablations with alternative corruption schemes or natural corpora?***
>
> To ensure that token-swapping perturbations reflect natural model failures, we measure the probability of the final answer token (e.g., the probability of "New York" in this case). After applying the perturbation, we retain only those examples where the model still assigns high probability to the (now incorrect) answer token. This indicates that the model is relying solely on contextual cues. Perturbations are selected from a broad pool of candidate replacements to satisfy this condition. In the memory-based setting, we ensure that the model assigns high probability to the correct answer stored in memory, while assigning low probability to the perturbed token. This setup ensures that the model's response is primarily influenced by memory rather than the current context. Moreover, our extensive ablations (e.g., Fig. 3) show that the circuits extracted with this probe dataset design is robust.

---

> > ### Author Response · Authors · 2025-05-30
> > **Further Response**
> >
> > ***(Q2): "Insufficient Generality: the method and probe dataset focuses only on extractive single-hop QA, while performance drops significantly on multi-hop reasoning tasks. It could be due design of probe dataset, but it is not clear whether the circuit could be extended to the general combination of extraction + multi-hop reasoning:  ……….You note poorer performance on multi-hop QA. Have you attempted to build a multi-hop probe dataset, and does that yield a distinct circuit?***:
> >
> >  We acknowledge that the multi-hop QA results in our paper are slightly weaker, though still within an acceptable range. Notably, we found it insightful that the single-hop QA circuit generalizes reasonably well to multi-hop QA scenarios, with only a modest drop in performance. This suggests that while the core mechanisms overlap, multi-hop QA likely involves additional components responsible for more complex forms of reasoning. Eliciting such a circuit would require a fundamentally different probe dataset specifically designed to capture these additional elements. We believe this direction merits a dedicated investigation and could form the basis of a future paper. We will clarify these distinctions in the camera-ready version to ensure a clearer separation between the single-hop and multi-hop setups.
> >
> >
> > ***"Line 245: Fig(R) is a reference to the section, not a figure:"***:
> > We thank the reviewer for bring this to our notice. We will update Fig. with Sec. in the final camera-ready version of the paper.
> >
> >
> > ***"Fig. 4 is a bit misleading. I believe, there is some typo, because there are two different visualisations of head [18, 30]. Maybe one of them should be [24, 8]?"***.
> > In Fig. (4), we show two instances of different examples where [18,30] is activated, as this head is one of the driver heads of the circuit in Vicuna.

---

> > > ### Comment · Reviewer_z9as · 2025-06-04
> > >
> > > Thank you for the detailed and thoughtful response. I especially appreciate the additional results on LLaMA-3-70B, which clearly demonstrate the scalability and effectiveness of AttnAttrib on larger models.
> > >
> > > Most of concerns are addressed, but the interaction between the context-faithfulness and memory-faithfulness circuits remains somewhat unclear. While the appendix suggests they operate independently, it would strengthen the paper to include more quantitative analysis - e.g., statistics on attention head or path overlap, or a direct comparison of the effects of ablating each circuit individually on extractive QA datasets (e.g., those in Fig. 3). This could clarify whether the circuits are independent. Could you provide such analysis?

---

> > > > ### Author Response · Authors · 2025-06-04
> > > > **Response**
> > > >
> > > > We are glad to know that most of your concerns have been addressed!
> > > >
> > > > Based on your suggestion, we will add more analyses on the comparison between the two kinds of circuits.  We will run these new suggested experiments for the camera-ready version in the next few days. We will also try to report these results in the discussion here before the discussion period ends. Many thanks for the suggestion!

---

### Official Review · Reviewer_F4x8 · 2025-05-12

**Rating:** 7
**Confidence:** 4
**Ethics Flag:** 1

**Summary:**

This paper investigates mechanistic circuits in LLMs for extractive QA, distinguishing between context-faithful and memory-faithful circuits. It introduces AttnAttrib, a efficient method for data attribution which obtains state-of-the-art attribution results across various extractive QA benchmarks. This paper further shows that the attributions from AttnAttrib can be used towards improving generalization in extractive QA tasks by steering the model towards context faithfulness.

**Reasons To Accept:**

1. The extraction of context- and memory-faithful circuits provides a principled framework for understanding how LLMs balance parametric knowledge and contextual information. The observation that distinct circuits govern these behaviors is compelling and empirically validated.

2. The authors validate their circuits across multiple models (Vicuna, LLaMA-3, Phi-3), datasets (HotPotQA, Natural-Questions, NQ-Swap), and even test generalization to LLaMA-3-70B.

**Reasons To Reject:**

1. The quality and representativeness of the circuits heavily rely on the designed probe datasets. Although carefully constructed, the generalizability to more diverse, noisy, or longer-context settings could be better discussed.

2. While ATTNATTRIB is efficient at inference time, the initial circuit extraction (causal analysis, patching) appears computationally expensive.

---

> ### Author Response · Authors · 2025-05-30
> **Response to Reviewer**
>
> We thank the reviewer for finding our work insightful in understanding the inner mechanisms of extractive QA in language models. We are glad that the reviewer find that our experiments have a wide-coverage in terms of models and datasets. Below we answer the questions raised by the reviewer:
>
> ***"The quality and representativeness of the circuits heavily rely on the designed probe datasets. Although carefully constructed, the generalizability to more diverse, noisy, or longer-context settings could be better discussed"*** :
>
> The reviewer raises an important point. To obtain robust mechanistic insights, we used a controlled and well-designed probe dataset to ensure the extraction of "clean" circuits. This practice is standard in the mechanistic interpretability community and is followed by several influential works. To assess whether the extracted circuit generalizes to more diverse and noisy datasets, we ablate the circuit—originally extracted from the probe dataset—across multiple popular QA benchmarks, as shown in Fig. 3. Despite being derived from a controlled setup, the circuit remains valid across noisier datasets.
> To further test generalization to long-context QA scenarios, we designed an additional experiment during the rebuttal period. We constructed long-context inputs by merging documents from HotPotQA and evaluated the circuit’s impact when ablated across 200 examples. The results are as follows:
>
> 	• Doc Length: 10 — Initial Accuracy: 74%, Ablated Circuit: 3%, Random Circuit: 71%
>
> 	• Doc Length: 20 — Initial Accuracy: 69%, Ablated Circuit: 2%, Random Circuit: 68%
>
> 	• Doc Length: 30 — Initial Accuracy: 67%, Ablated Circuit: 2.5%, Random Circuit: 66%
>
> These results demonstrate that the extracted circuit continues to capture core mechanisms even in long-context settings. We will include this new experiment in the camera-ready version of the paper.
>
>
> ***"While ATTNATTRIB is efficient at inference time, the initial circuit extraction (causal analysis, patching) appears computationally expensive."*** :
>
> We agree with the reviewer’s point about the initial circuit extraction being expensive. However, we note that the initial circuit extraction is a one-time cost, which can be especially advantageous when attributions are served alongside answers to many end-users, as it significantly reduces inference-time costs at scale.

---

> > ### Comment · Reviewer_F4x8 · 2025-06-09
> >
> > Thanks for you reply. I will maintain my positive rating.

---

### Official Review · Reviewer_mTR6 · 2025-05-12

**Rating:** 7
**Confidence:** 4
**Ethics Flag:** 1

**Summary:**

This paper investigates the mechanisms underlying extractive question answering using mechanistic interpretability toolkit. The authors identify distinct circuits responsible for answering questions from memory and from context. They also find the attention head most critical for answering from context and demonstrate that its attention pattern can be used to find more faithful answers. The experiments are conducted on the LLaMA-3, Vicuna, and Phi-3 language models.

**Questions To Authors:**

- What is the exact difference between the "a" and "m" nodes in Figure 1?
- While you describe how questions and contexts are "corrupted" in Section 3.1, it remains unclear whether the same corruption method is used in Section 3.2 or other parts of the paper. Are there any differences in the corruption procedures across different experiments?
- Could you please elaborate on Algorithm 1, particularly the GetMaxSpan function? This function appears to be central to the algorithm, yet it is not mentioned elsewhere in the paper, leaving its implementation unclear.
- From what I understood, the nodes (either MLPs or attention heads) were patched using their activations on "corrupted" questions and contexts. Did you explore using alternative values for patching these nodes? Different patching values may lead to extracting different circuits (see Transformer Circuit Faithfulness Metrics are not Robust by Miller et al.).
- In Appendix A.2, you highlight that head (17, 24) in LLaMA 3 is effective for extracting the correct attention span. Interestingly, the same head appears in the top-right region of Figure 10(b) in Listening to the Wise Few: Select-and-Copy Attention Heads for Multiple-Choice QA, where it is identified as highly effective for Multi-Choice Question Answer extraction via Query-Key interactions. I also noticed considerable overlap between the heads shown in that figure and those listed in Section D.2 ("LLaMA-3-8B") of your paper. Could you investigate this further? Specifically, how frequently are heads that are useful for Multiple-Choice Question Answering also effective for Extractive Question Answering?

**Reasons To Accept:**

- The paper contributes to the interpretability field by investigating the inner mechanisms of relatively recent language models.
- It demonstrates that the extractive question answering capabilities of current models can be improved by using insights gained from understanding their internal workings.

**Reasons To Reject:**

- The main problem: the paper is difficult to follow; many details, particularly those related to the methodology, remain unclear to me after reading (**see the "Questions" section**).
- The paper relies too heavily on the appendix, which contributes to its lack of clarity and makes it harder to understand the main content.
- The manuscript would benefit from proofreading, as it contains typos. Additionally, the font size in many figures is extremely small, making them very difficult to read.
- The concept of AttnAttrib (Section 4.1) appears closely related to the AttnScore mechanism introduced in Listening to the Wise Few: Select-and-Copy Attention Heads for Multiple-Choice QA by Tulchinskii et al. However, the paper does not cite this prior work or explore the relationship between the two approaches (see also "Questions" section).

**UPD: The authors clarified previously unclear aspects of the paper, committed to including these clarifications in the camera-ready version, and provided additional results and explanations. As a result, I have raised my final score.**

---

> ### Author Response · Authors · 2025-05-30
> **Response to Reviewer**
>
> We thank the reviewer for the extensive review and acknowledging that our paper contributes to the mechanistic interpretability field of language models with various actionable insights about the extractive QA task.
>
> Below we address the reviewer's comments:
>
> ***What is the exact difference between the "a" and "m" nodes in Figure 1?*** : a refers to the attention layers whereas m refers to the MLP layers. We will add a legend corresponding to this in the final camera-ready version of the paper.
>
> ***While you describe how questions and contexts are "corrupted" in Section 3.1, it remains unclear whether the same corruption method is used in Section 3.2 or other parts of the paper. Are there any differences in the corruption procedures across different experiments?*** :  The corruptions are selected from a large pool of tokens, such that the probability of the answer from the "clean" model drops significantly.  In terms of implementations, we implement an iterative function which scans over a set of tokens and selects the token such that the probability drop from the "clean" instance of the model is maximized.  Across different experiments, this procedure is used. We will highlight this better in the final camera-ready version of the paper.
>
> ***"Could you please elaborate on Algorithm 1, particularly the GetMaxSpan function? This function appears to be central to the algorithm, yet it is not mentioned elsewhere in the paper, leaving its implementation unclear."*** :  The GetMaxSpan function utilizes the main attention head from the context-faithfulness circuit and extracts the sentence (amongst many in the context) which contains the token with the highest attention from that attention head. We have explained this in Line 276-277,  but will make it clearer in the final camera-ready version based on the reviewer's suggestion.
>
>
> ***"The paper relies too heavily on the appendix, which contributes to its lack of clarity and makes it harder to understand the main content."*** :The reviewer raises a valid concern. However, we note that mechanistic interpretability papers—particularly those focused on circuits—often involve highly detailed analyses that cannot be fully included in the main paper due to space constraints. Notably, prior works on circuits (e.g., 1, 2) also include extensive appendices for this reason. Similarly, our work relies on the Appendix to elaborate on technical details essential for full reproducibility. That said, we have made a concerted effort to present the key findings and takeaways clearly in the main paper.
>
>
> ***" The manuscript would benefit from proofreading, as it contains typos. Additionally, the font size in many figures is extremely small, making them very difficult to read."***  : We will increase the resolution in the figures in the camera-ready so that they are better readable. Thanks for the suggestion!
>
>
> ***"From what I understood, the nodes (either MLPs or attention heads) were patched using their activations on "corrupted" questions and contexts. Did you explore using alternative values for patching these nodes? Different patching values may lead to extracting different circuits (see Transformer Circuit Faithfulness Metrics are not Robust by Miller et al.)."*** : In our experiments, we applied different corruption instances prior to the patching operation and found the extracted circuits to be consistent. This indicates that, for extractive QA tasks, as long as the corrupted run yields a low probability for the correct answer (as compared to the clean instance), the resulting circuits remain similar. We will include this clarification in the camera-ready version. Additionally, we present extended ablations (e.g., Fig. 3) demonstrating that the circuits obtained through our patching procedure are robust—ablating them results in a substantial drop in extractive QA accuracy.

---

> > ### Author Response · Authors · 2025-05-30
> > **Further Response**
> >
> > ***"In Appendix A.2, you highlight that head (17, 24) in LLaMA 3 is effective for extracting the correct attention span. Interestingly, the same head appears in the top-right region of Figure 10(b) in Listening to the Wise Few: Select-and-Copy Attention Heads for Multiple-Choice QA, where it is identified as highly effective for Multi-Choice Question Answer extraction via Query-Key interactions. I also noticed considerable overlap between the heads shown in that figure and those listed in Section D.2 ("LLaMA-3-8B") of your paper. Could you investigate this further? Specifically, how frequently are heads that are useful for Multiple-Choice Question Answering also effective for Extractive Question Answering?"***:   We thank the reviewer for bringing this interesting paper to our attention. While multi-choice QA differs from extractive QA—primarily due to the additional step of selecting the correct option—the two tasks share the underlying requirement of identifying relevant context to answer the question. To assess whether our context-faithfulness circuit generalizes to multi-choice QA, we performed an ablation study on a subset of Cosmos-QA examples using LLaMA-3-8B. The results are as follows:
> >
> > {Original Acc: 0.543,  Context-Faithfulness Circuit Ablation: 0.291,  Random Circuit: 0.52}
> >
> > Although there is a noticeable drop in performance, it is less pronounced than in the extractive QA setting (e.g., Fig. 3 in our paper). This suggests that components of the extractive QA circuit contribute meaningfully to multi-choice QA, though additional specialized components may also be involved in solving that task. This warrants further investigation and requires designing a new probing dataset to elucidate the full circuit.

---

> > > ### Comment · Reviewer_mTR6 · 2025-06-01
> > >
> > > > We thank the reviewer for bringing this interesting paper to our attention. While multi-choice QA differs from extractive QA—primarily due to the additional step of selecting the correct option—the two tasks share the underlying requirement of identifying relevant context to answer the question. To assess whether our context-faithfulness circuit generalizes to multi-choice QA, we performed an ablation study on a subset of Cosmos-QA examples using LLaMA-3-8B. The results are as follows:
> > > {Original Acc: 0.543, Context-Faithfulness Circuit Ablation: 0.291, Random Circuit: 0.52}
> > > Although there is a noticeable drop in performance, it is less pronounced than in the extractive QA setting (e.g., Fig. 3 in our paper). This suggests that components of the extractive QA circuit contribute meaningfully to multi-choice QA, though additional specialized components may also be involved in solving that task. This warrants further investigation and requires designing a new probing dataset to elucidate the full circuit.
> > >
> > > It would be valuable to include these results and the accompanying discussion in the paper - perhaps in the Appendix.

---

> > > > ### Author Response · Authors · 2025-06-01
> > > > **Response**
> > > >
> > > > Thanks, we will include these results in the Appendix of the final version!

---

> > ### Comment · Reviewer_mTR6 · 2025-06-01
> >
> > > The GetMaxSpan function utilizes the main attention head from the context-faithfulness circuit and extracts the sentence (amongst many in the context) which contains the token with the highest attention from that attention head. We have explained this in Line 276-277, but will make it clearer in the final camera-ready version based on the reviewer's suggestion.
> >
> > But what is the exact definition of this? There are many possible ways to “extract the sentence (among many in the context) that contains the token with the highest attention from that attention head.” Do you simply sum the attention values for the tokens in each sentence and compare these sums across sentences? Do you use only the top-1 sentence based on this sum, or the top-2? Do you choose a fixed number of top sentences, or is the number determined dynamically? Or are you using a completely different approach? Lines 276-277 are super vague and describe only the general idea.

---

> > > ### Author Response · Authors · 2025-06-01
> > > **Response**
> > >
> > > Given a context C consisting of N sentences (e.g., c1, c2, ..., cN) and an output of L tokens, GetMaxSpan selects, for each generated token, the sentence containing the context token that receives the highest attention from a designated attention head in the circuit at the time of generation. For example, when the first output token is generated, c2 might be selected; for the second token, c7 might be selected. The score assigned to each selected sentence (e.g., c2, c7) is the normalized attention value of the most attended token in that sentence (from the attention head of the circuit), as recorded in the append step of the algorithm. Therefore instead of aggregating attention across all tokens in a sentence, the method compares sentences based on their single highest attention value.
> > >
> > > We hope this clarifies your question. We will add this detail to the camera-ready version for better readability!

---

> > > > ### Comment · Reviewer_mTR6 · 2025-06-02
> > > >
> > > > Yes, it is clear now, thank you for explanation

---

> > ### Comment · Reviewer_mTR6 · 2025-06-01
> >
> > > In our experiments, we applied different corruption instances prior to the patching operation and found the extracted circuits to be consistent. This indicates that, for extractive QA tasks, as long as the corrupted run yields a low probability for the correct answer (as compared to the clean instance), the resulting circuits remain similar. We will include this clarification in the camera-ready version. Additionally, we present extended ablations (e.g., Fig. 3) demonstrating that the circuits obtained through our patching procedure are robust—ablating them results in a substantial drop in extractive QA accuracy.
> >
> > It would be good to see the results of the particular experiments of comparison of circuits, obtained by different patching methods. Feel free to share the results in the current discussion.

---

> > ### Author Response · Authors · 2025-06-01
> > **Response**
> >
> > The probe dataset used in our study includes annotations for both the subject and answer tokens within the context. To generate corrupted instances, we perturb the subject and answer tokens in the context as well as the subject token in the question. In the current version of the paper, we apply a replacement-based corruption strategy, where subject and answer tokens are substituted with alternative entities (e.g., Space Needle with Big Sur, New York with California). During experimentation, we also explored an alternative corruption method by injecting small Gaussian noise into the input embeddings of the subject and answer tokens. We found that the circuits extracted using this noise-based perturbation closely resembled those from the replacement method.
> >
> > Below are the extracted circuits corresponding to each corruption strategy (for Llama-8B):
> >
> > Circuit with replacement-based corruption:
> >
> > 	• Attention Layers: [27, 23, 31, 24, 25, 29, 21, 30]
> >
> > 	• Attention Heads: [[27, 20], [23, 27], [31, 7], [17, 24], [25, 12], [31, 20], [24, 27], [27, 6], [26, 13], [16, 1], [31, 6], [29, 31], [31, 3], [30, 12]]
> >
> > Circuit with Gaussian noise-based corruption:
> >
> > 	• Attention Layers: [27, 23, 31, 25, 24, 29, 21, 30]
> >
> > 	• Attention Heads: [[27, 20], [23, 27], [31, 7], [17, 24], [25, 12], [31, 20], [24, 27], [27, 6], [26, 13], [16, 1], [31, 6], [29, 31], [30, 12], [31, 3]]
> >
> > As shown, the circuit components related to context faithfulness remain largely consistent across both corruption strategies, with only minor variations in the ordering. We will add this discussion in the Appendix of the camera-ready version.

---

> > > ### Comment · Reviewer_mTR6 · 2025-06-02
> > >
> > > These are interesting results. I have raised my final score based on your explanations and the additional experiments. For the camera-ready version, I suggest including a comparison with other types of patching (e.g., replacing activations of attention heads or MLP layers with zeros, average values, etc.) to further strengthen the analysis.

---

> > > > ### Author Response · Authors · 2025-06-04
> > > > **Response**
> > > >
> > > > Thank you! Based on your suggestion, we will add these analyses to the final camera-ready version. Thanks for the suggestion again!

---

### Decision · Program_Chairs · 2025-07-08

**Decision:**

Accept

**Comment:**

The paper combines a novel approach to data attribution using causal mediation analysis, with convincing empirical results, showing improvements over multiple baseline methods, with interesting practical applications. The reviewers all agree that the paper should be accepted.